# GSH and Zinc Supplementation Attenuate Cadmium-Induced Cellular Stress and Stimulation of Choline Uptake in Cultured Neonatal Rat Choroid Plexus Epithelia

**DOI:** 10.3390/ijms22168857

**Published:** 2021-08-17

**Authors:** Samantha D. Francis Stuart, Alice R. Villalobos

**Affiliations:** 1Department of Nutrition and Food Science, Texas A&M University, College Station, TX 77843, USA; sdfstuart@gmail.com; 2Department of Veterinary Integrative Biosciences, Texas A&M University, College Station, TX 77843, USA

**Keywords:** choroid plexus, cadmium, Zinc, GSH, choline, transport, cellular stress, oxidative stress

## Abstract

Choroid plexus (CP) sequesters cadmium and other metals, protecting the brain from these neurotoxins. These metals can induce cellular stress and modulate homeostatic functions of CP, such as solute transport. We previously showed in primary cultured neonatal rat CP epithelial cells (CPECs) that cadmium induced cellular stress and stimulated choline uptake at the apical membrane, which interfaces with cerebrospinal fluid in situ. Here, in CPECs, we characterized the roles of glutathione (GSH) and Zinc supplementation in the adaptive stress response to cadmium. Cadmium increased GSH and decreased the reduced GSH-to-oxidized GSH (GSSG) ratio. Heat shock protein-70 (Hsp70), heme oxygenase (HO-1), and metallothionein (Mt-1) were induced along with the catalytic and modifier subunits of glutamate cysteine ligase (GCL), the rate-limiting enzyme in GSH synthesis. Inhibition of GCL by *l*-buthionine sulfoximine (BSO) enhanced stress protein induction and stimulation of choline uptake by cadmium. Zinc alone did not induce Hsp70, HO-1, or GCL subunits, or modulate choline uptake. Zinc supplementation during cadmium exposure attenuated stress protein induction and stimulation of choline uptake; this effect persisted despite inhibition of GSH synthesis. These data indicated up-regulation of GSH synthesis promotes adaptation to cadmium-induced cellular stress in CP, but Zinc may confer cytoprotection independent of GSH.

## 1. Introduction

The choroid plexus (CP) epithelia form the blood–cerebrospinal fluid (CSF) barrier. These transporting epithelia secrete CSF and selectively exchange inorganic and organic solute between blood and CSF, thereby regulating fluid/electrolyte balance, nutrient availability, and metabolite and xenobiotic clearance in CSF. CSF is continuous with the extracellular fluid surrounding neurons and glia in the brain. Thus, central neural responses to neurotransmitters and neuropeptides are contingent upon the capacity of CP to maintain CSF homeostasis [1,2,3,4]. Modulation of solute transport by physicochemical stressors, such as heavy metals, may compromise regulation of CSF volume and composition and disrupt central neural homeostasis.

The CP sequesters heavy metals and protects the brain from these neurotoxins [5,6], potentially subjecting itself to modulation or injury by these very same agents. Metals sequestered by CP include cadmium [7,8,9,10], a toxic contaminant metal with a biological half-life of 15–30 years [11]. Given the naturally high cadmium content in tobacco plants, cigarette smoking is the predominant means of human exposure. Nonetheless, cadmium has broad industrial applications and is found in starchy root plants, mollusks, and other natural food sources [11,12]. Cadmium is a non-Fenton metal that can induce oxidative stress by indirectly increasing intracellular production or accumulation of free radicals and reactive species [13]. Cadmium disrupts cellular redox balance by displacing Fenton metals iron and copper, inhibiting complex III in the electron-transport chain, and depleting the intracellular antioxidant glutathione [14,15,16]. In mice, chronic cadmium exposure elicited dose-dependent changes indicative of cellular injury in CP epithelia, such as shortening and loss of microvilli and intracellular vacuole formation [10]. Investigation of the potential effects of cadmium on function, such as solute transport, is limited. We previously showed in primary cultured neonatal rat CP epithelial cells (CPECs) that subchronic exposure to 500 nM cadmium induced an oxidative cellular stress response and stimulated apical uptake of choline [17]. Choline is a model substrate for organic cation transporter-2 (OCT-2, SLC22A2) localized to the apical membrane [18] and precursor to the neurotransmitter acetylcholine (ACh). Choline is critical for normal brain development and cognitive functions, such as learning and memory, in neonatal animals [19,20,21]. Stimulation of choline uptake at the apical membrane of CPECs is analogous to increased removal of choline from the CSF compartment in situ. This could potentially limit central availability of choline and ACh. CP epithelia sequester cadmium and other metals, but mechanisms that might minimize stress modulation of solute transport and other functions critical to CNS homeostasis are not fully characterized.

Glutathione (GSH) is the most abundant intracellular thiol and predominant intracellular antioxidant [22]. GSH is readily available and can directly bind cadmium. As such, it is considered the first line of defense against cadmium-induced oxidative cellular stress [23,24]. However, cadmium can deplete intracellular GSH [13,25]. We showed in CPECs that a precursor to GSH, *n*-acetylcysteine, attenuated induction of Hsp70 and stimulation of apical choline uptake by cadmium. Conversely, *l*-buthionine-sulfoximine (BSO), an inhibitor of the rate-limiting enzyme in GSH synthesis glutamate cysteine ligase (GCL), enhanced both Hsp70 induction and stimulation of choline uptake [17]. Thus, a cellular stress response was mounted in response to oxidative stress induced by cadmium. This suggested GSH might be critical to the epithelium’s adaptation to cadmium-induced cellular stress. GCL is highly expressed in CP, and GSH synthesis is integral to γ-glutamyl cycling and amino acid transport [26], as well as phase II metabolism and drug clearance from CSF [27]. In CP, the antioxidant GSH system may serve to regulate accumulation of reactive oxygen species (ROS) produced by basal metabolism and ROS generated upon induction of oxidative stress, such as by elevated levels of H_2_O_2_ in CSF [28]. However, the role of GSH in adaptation to cellular stress elicited by cadmium or other contaminant metals sequestered by CP has not been fully elucidated.

Zinc, an essential mineral normally accumulated by CP [29,30,31], is a co-factor and structural stabilizer for enzymes, including carbonic anhydrase, which is central to CSF production [32,33]. Furthermore, Zinc is a redox inert metal that promotes cellular antioxidant mechanisms, protecting sulfhydryl groups from free radicals, and abridging reactive species formation through antagonism of redox-active metals; e.g., iron and copper [34]. In neurons, Zinc supplementation abates cadmium-induced oxidative cellular stress [35], and Zinc availability may regulate GSH metabolism [34,36,37]. In mice, co-treatment with Zinc protected against cadmium-induced oxidative stress in the liver, maintained GSH at levels comparable to control, and attenuated lipid peroxidation [38]. Zinc also increased expression of metallothionein (MT), which can bind cadmium and minimize subsequent disruption of cell biology and cytotoxicity [39]. Nonetheless, in MT-null mice, Zinc was protective against cadmium-induced nephrotoxicity in a GSH-dependent manner [40]. Despite marked accumulation in CP, the capacity of Zinc to minimize cellular injury by contaminant metals or other stressors has not been investigated. Given the critical role of CP in brain homeostasis and global prevalence of deficiency in Zinc intake [41,42], elucidating the role of Zinc in CP biology is warranted.

Using primary cultured neonatal rat CPECs and isolated CP tissues exposed to sub-micromolar cadmium, our aim was to characterize the potentially protective roles of GSH and Zinc supplementation in the adaptive stress response to cadmium.

## 2. Results

### 2.1. Cultured Choroid Plexus Epitheial Cells (CPECs) Accumulate Cadmium (Cd) and Zinc (Zn)

Experimental treatments of CPECs and isolated CP tissues implemented in this study are described in detail in the Materials and Methods (Section 4.4) and outlined in the schematic diagram in Figure 1. Nonetheless, reporting of experimental results will include succinct descriptions of the respective treatments and references to Figure 1. The protocol used in this study to expose cells to 500 nM CdCl_2_ was similar to that implemented in our prior study of cadmium-induced effects in CPECs [17]. In the present study, we investigated whether Zinc supplementation might alter cadmium-induced responses in CPECs under conditions in which Zinc was present both before and during cadmium exposure.

We previously showed that CPECs exposed for 12 h to 500 nM CdCl_2_ in serum-free medium (SFM) accumulated cadmium (*Cd*) [17], which was consistent with accumulation of Cd in the choroid plexus of cadmium-exposed rodents [8,9,10]. In this study, we sought to determine whether CPECs supplemented with Zinc would indeed accumulate Zinc, as reported for the choroid plexus of intact rodents [29,30,31]. Furthermore, if Zinc were found to alter cadmium-induced effects in CPECs, this could be due in part to decreased cellular accumulation of cadmium. Therefore, we also sought to compare accumulation of Zinc and cadmium in CPECs exposed to combinations of Zinc and cadmium. For this and subsequent experiments, CPECs were maintained in maintenance medium until initiation of experimental treatment. Briefly, CPECs were first incubated for 48 h in maintenance medium without Zinc, to yield what will be referred to as non-supplemented cells, or in maintenance medium with 25 µM ZnCl_2_, to yield what will be referred to as Zn-supplemented cells. Non-supplemented cells were then pre-treated in Zinc-free SFM for 12 h and divided into two sets: one set was incubated for 12 h in Zinc-free SFM with 0 CdCl_2_ (Control), while the other set was exposed for 12 h to 500 nM CdCl_2_ in Zinc-free SFM (Cadmium, Cd). In parallel, Zn-supplemented cells were pre-treated in SFM with 10 µM ZnCl_2_ and then divided into two sets: one set was incubated for 12 h in fresh SFM with 10 µM ZnCl_2_ (Zinc, Zn), while the other was exposed for 12 h to 500 nM CdCl_2_ in SFM with 10 ZnCl_2_ (Zn + Cd) (Figure 1A).

Cellular content of Zinc and cadmium was measured by inductively coupled plasma mass spectrophotometry (ICP-MS) and normalized to cell protein (Table 1). In control CPECs, mean content of Zinc was 333.82 ng/mg protein, and that of cadmium was 1.02 ng/mg protein. 12 h cadmium exposure increased cellular cadmium content to 265.79 ng/mg protein, but Zinc content remained similar to control (348.99 ng/mg protein). Zinc supplementation increased Zinc content to 556.61 ng/mg protein (*p* < 0.01 vs. Control), while cadmium content remained similar to control (1.14 ng/mg protein). After concurrent exposure to Zinc and cadmium, Zinc content was slightly less than after exposure to Zinc alone (*p* < 0.10, Zn + Cd vs. Zn); cadmium content was similar to that in cells exposed to only cadmium (*p* > 0.54; Zn + Cd vs. Cd). Thus, CPECs did accumulate Zinc and cadmium and under these experimental conditions, Zinc supplementation did not attenuate cadmium accumulation, whereas cadmium exposure modestly decreased Zinc accumulation.

### 2.2. Cadmium Alters Intracellular Glutathione (GSH) and Glutathione Sulfide (GSSG) in CPECs

Cadmium can deplete intracellular GSH and decrease the ratio of GSG to GSSG, which is an index of oxidized redox potential [22]. Thus, we examined whether 12 h exposure to 500 nM CdCl_2_ might alter GSH chemistry, and whether Zinc supplementation might modify any observed changes. We followed the same treatment protocol used to examine cellular Cd and Zn accumulation (Table 1). Non-Zinc supplemented cells were pre-treated (12 h) in SFM and then divided into two sets: one was exposed for another 12 h in SFM with 0 CdCl_2_ (Control), while the other was exposed for 12 h to 500 nM CdCl_2_ in SFM (Cd). In parallel, Zn-supplemented cells were pre-treated (12 h) in SFM with 10 µM ZnCl_2_ and then divided into two sets: one was incubated for another 12 h in SFM with 10 µM ZnCl_2_ (Zn), while the other was exposed for 12 h to 500 nM CdCl_2_ in SFM with 10 µM ZnCl_2_ (Zn + Cd), (Figure 1A). We then analyzed GSH and GSSG concentrations by luminescence and calculated the ratio of GSH to GSSG (GSH:GSSG ratio; Table 2). Compared to control, cadmium exposure increased GSH and GSSG concentrations 1.75- and 7-fold (*p* < 0.05), respectively. Mean GSH:GSSG ratio in controls was 320, but was 45 in cadmium-exposed cells (*p* < 0.05). Zinc alone did not alter GSH or GSSG concentrations or GSH:GSSG ratio. Zinc also did not abate cadmium-elicited increases in GSH or GSSG or decreases in GSH:GSSG ratio. Cadmium elicited increases in GSH, suggesting that GSH synthesis was up-regulated.

### 2.3. Cadmium Induced Glutamate Cysteine Ligase (GCL) Subunits and Stress Proteins in CPECs

Increases in GSH elicited by cadmium suggested that GSH synthesis was up-regulated. This might be a component of the collective adaptation to the cellular oxidative stress elicited by cadmium. GCL catalyzes the rate-limiting step in GSH synthesis, which is a critical regulatory point in de novo synthesis of GSH [22,43]. A catalytic subunit, GCLC, and modifier subunit, GCLM, comprise GCL. We sought to determine whether GCLC and GCLM were up-regulated within the context of the cellular stress response mounted in upon exposure to cadmium in CPECs. Thus, we analyzed time-dependent induction of GCLC and GCLM, along with heat shock protein-70 (Hsp70), heme oxygenase (HO-1), and metallothionein in non-Zinc supplemented CPECs. To this end, non-supplemented CPECs were pre-treated in SFM (12 h) and then exposed to 0 (Control) or 500 nM CdCl_2_ (Cd) in SFM for 12 h (Figure 1B). At 3, 6, 9, and 12 h, representative cadmium-exposed and time-matched control cells were collected for analysis of mRNA expression for the genes encoding for GCL subunits and respective stress proteins by quantitative real-time polymerase chain reaction (qRT-PCR), and for evaluation of protein expression of GCL subunits, Hsp70, and HO-1 by immunoblot analyses.

Cadmium induced mRNA expression for rat genes encoding for heat shock protein-70 (Hspa4), heme oxygenase (Hmox1), and metallothionein (Mt1) in a time-dependent manner (Figure 2A). Hspa4 mRNA was induced 7- and 9-fold at 6 h and 9 h, respectively, and was still 4-fold greater than control at 12 h. Hmox1 mRNA was induced 8-fold at 3 h and further increased to 70-fold at 12 h. Similarly, Mt1 mRNA was induced 5-fold at 3 h and 55-fold at 12 h. Cadmium treatment also induced GCL subunit gene expression (Figure 2B). The mRNA expression for the gene encoding for GCLC (Gclc) was induced 3-fold at 3 h and induced 5- and 7.5-fold at 9 h and 12 h, respectively. The mRNA expression for the gene encoding for GCLM (Gclm) was induced 1.5-fold at 3 h, but increased to 9-fold at 6 h and induced 11.5-fold at 9 h; expression was still 8-fold greater than control at 12 h.

Protein expression of Hsp70, HO-1, and GCLM was also induced by cadmium (Figure 3). Hsp70 was induced 3-fold at 6 h through 9 h and remained elevated (4.5-fold) at 12 h. HO-1 was induced 2.5-fold at 3 h and increased incrementally to 12-fold greater than control at 12 h (Figure 3A). GCLC protein expression was similar to control for the 12 h duration of cadmium exposure (*p* > 0.50), whereas GCLM protein was induced 2-fold after 12 h of exposure (Figure 3B).

Induction of stress protein and GCL subunit gene expression was examined in isolated neonatal CP to determine whether a similar response could be elicited in the intact tissues. Lateral and fourth CP tissues isolated from neonatal rats and incubated with 0 (Control) or 500 nM CdCl_2_ (Cadmium) in SFM for 24 h (Figure 1C). Given the complex organization of epithelial cells and vascular tissues (endothelial, smooth muscle) of the intact choroid plexus versus the monolayer of epithelial cells in CPECs, we extended the exposure time to 24 h versus the 12 h exposure time for single-layered CPECs. In cadmium-treated tissues, mRNA expression of Hspa4, Hmox1, Gclc, and Gclm were twice those in controls; Mt1 gene expression was 9 times that in controls (*p* < 0.05, Figure 4).

### 2.4. Zinc Attenuates Increases in Apical Choline Uptake in Cadmium-Exposed CPECs

We assessed to what extent treatment with Zinc alone might alter apical choline uptake. CEPCs were incubated in maintenance medium not supplemented with ZnCl_2_ (i.e., non-supplemented cells) and then treated (37 °C) for 24 h in SFM containing 0 (Control) or 5–100 µM ZnCl_2_ (Figure 1D). After experimental treatment, we assayed 30 min apical uptake of 10 µM [^3^H]choline ± 750 µM hemicholinium-3 (HC-3) in Zinc-free artificial cerebrospinal fluid (aCSF) at 37 °C (Table 3). Thirty-minute choline uptake was 3274.27 ± SE 287.13 pmol/mg protein in controls (*n* = 3). Pre-treatment with 5–50 µM ZnCl_2_ did not alter choline uptake (*p* > 0.72), but treatment with 100 µM ZnCl_2_ decreased uptake by 90% (*p* < 0.05 vs. Control).

Subsequently, we examined the potential for Zinc supplementation to disrupt cadmium-elicited stimulation of apical choline uptake in CPECs by comparing choline uptake in non-supplemented cells exposed for 12 h to 500 nM CdCl_2_ to that in Zn-supplemented cells exposed for 12 h to 500 nM CdCl_2_ (Figure 1A). We followed the same treatment protocol used to examine cellular Cd and Zn accumulation (Table 1) and GSH chemistry (Table 2). Non-Zinc supplemented cells were pre-treated (12 h) in SFM and divided into two sets: one was incubated for 12 h in SFM (Control), while the other was exposed for 12 h to 500 nM CdCl_2_ in SFM (Cd). In parallel, Zn-supplemented cells were pre-treated (12 h) in SFM with 10 µM ZnCl_2_ and divided into two sets: one was incubated for 12 h in SFM with 10 µM ZnCl_2_ (Zn), while the other was exposed for 12 h to 500 nM CdCl_2_ in SFM with 10 µM ZnCl_2_ (Zn + Cd) (Figure 1A). Following experimental treatment, 30 min apical uptake of 10 µM [^3^H]choline (± 750 µM HC-3) was assayed in aCSF free of cadmium and Zinc (Figure 5). Compared to control, cadmium increased choline uptake by 60% (*p* < 0.05), but Zinc alone had no effect. However, Zinc supplementation abated stimulation of choline uptake by cadmium (*p* < 0.05, Zn + Cd vs. Cd), such that uptake was comparable to control (*p* > 0.34).

### 2.5. Zinc Supplementation and Inhibition of GSH Synthesis Modify Cellular Stress Responses and Apical Choline Uptake in Cadmium-Exposed CPECs

Based on increases in GSH and induction of GCL subunits elicited by cadmium, we reasoned that up-regulation of GSH synthesis facilitated adaptation to cadmium-induced cellular stress. Thus, we predicted that inhibiting GSH synthesis to remove this component of adaptation would lead to compensatory increases in induction of heat shock protein-70, heme oxygenase, and metallothionein. Nonetheless, Zinc supplementation did not modulate GSH synthesis or prevent decreases in GSH:GSSG ratio by cadmium, but did attenuate stimulation of apical choline uptake in cadmium-exposed CPECs. Our prior work indicated stimulation of apical choline uptake in CPECs was associated with induction of the cellular oxidative stress response [17]. Thus, we predicted Zinc supplementation would still diminish cadmium-elicited stimulation of choline uptake and stress protein induction, if GSH synthesis were inhibited. To test these predictions, we used *l*-buthionine sulfoximine (BSO) to inhibit GCL [44] in Zn-supplemented and non-supplemented CPECs exposed to cadmium. This would permit evaluation of the efficacy of Zinc to modify the cellular stress response and attenuate stimulation of apical choline uptake in cadmium-treated CPECs despite attenuation of GSH synthesis (Figure 1E).

For these experiments, CPECs were first incubated for 48 h in maintenance medium without Zinc; i.e., non-supplemented cells, or with 25 µM ZnCl_2_; i.e., Zn-supplemented cells. To test the effects of cadmium and Zinc supplementation under conditions in which GSH synthesis was inhibited before and during cadmium exposure, 100 µM BSO was added to SFM used for pre-treatment and cadmium exposure of non-supplemented and Zn-supplemented CPECs. In the absence of BSO, a group of non-supplemented cells was pre-treated (12 h) in SFM and then divided into two sets; one was incubated for 12 h in SFM without cadmium (Control), while the other was exposed for 12 h to 500 nM CdCl_2_ in SFM (Cd). In the absence of BSO as well, a group of Zn-supplemented cells was pre-treated (12h) in SFM with 10 µM ZnCl_2_ and then divided into two sets: one was incubated for 12 h with 10 µM ZnCl_2_ alone (Zn), while the other was exposed for 12 h to 500 nM CdCl_2_ in SFM with 10 µM ZnCl_2_ (Zn + Cd). After experimental treatments, cells were processed for analysis of GSH chemistry, gene expression, and apical choline uptake.

We assayed GSH and GSSG by luminescence and calculated the GSH:GSSG ratio (Table 4). Compared to control, cadmium increased GSH and GSSG concentrations (4.39 ± 0.727 µM vs. 9.35 ± 1.38 µM and 0.050 ± 0.017 vs. 0.167 ± 0.012 µM, respectively) and decreased the GSH:GSSG ratio (~232 vs. ~47; *p* < 0.05); however, Zinc alone had no effect. Zinc supplementation did not prevent the increases in GSH and GSSG concentrations or the decrease in GSH:GSSG ratio elicited by cadmium. BSO alone decreased GSH concentration 90%, without increasing GSSG; the GSH:GSSG ratio was decreased to 8.7 (*p* < 0.05 vs. Control). In the presence of BSO, treatments with cadmium or Zinc did not alter GSSG concentration or the GSH:GSSG ratio versus BSO alone. Cadmium alone increased GSH, but in the presence of BSO, cadmium decreased GSH by 50% versus BSO alone (*p* < 0.03). In presence of BSO, Zinc did not alter GSH concentration versus BSO alone. In Zinc-supplemented cells exposed to cadmium with BSO, GSH concentration was similar to that in non-supplemented cells treated with cadmium and BSO.

To examine whether gene expression of stress proteins and GCL subunits were possibly up-regulated by Zinc supplementation prior to cadmium exposure, mRNA was analyzed in representative non-supplemented and Zn-supplemented CPECs. Briefly, representative non-supplemented CPECs were treated in SFM for 12 h, but not subjected to further treatments with BSO, Zinc, or cadmium; in parallel, representative Zn-supplemented cells were treated in SFM with 10 µM ZnCl_2_ for 12 h, but not subjected to further treatment with BSO or cadmium (Figure 1E). Thereafter, Hspa4, Hmox1, Mt1, and GCL subunit (Gclc and Gclm) mRNA expression was analyzed by qRT-PCR in Zn-supplemented and time-matched control cells (Figure 6A). Prior to cadmium exposure, Zinc supplementation did not alter Hspa4, Hmox1, Gclc, or Gclm mRNA (*p* > 0.10 vs. Control), but increased Mt1 mRNA nearly 19-fold (*p* < 0.03 vs. Control). Stress protein and GCL subunit gene expression was also analyzed in non-supplemented and Zn-supplemented CPECs exposed to cadmium (12 h) without or with inhibition of GSH synthesis by BSO (Figure 6B,C). Cadmium alone induced Hspa4 mRNA 2.75-fold, whereas in Zn-supplemented cells, cadmium did not induce Hspa4 mRNA expression; Zinc supplementation itself did not induce Hspa4 mRNA. Though BSO alone did not induce Hspa4 mRNA, it did enhance Hspa4 mRNA induction by cadmium to 10 times that in control. In Zn-supplemented cells exposed to cadmium with BSO, Hspa4 mRNA was induced at levels similar to those in cells exposed to only cadmium. In Zinc-supplemented cells treated with BSO, Hspa4 mRNA levels were similar to non-treated controls.

Hmox1 mRNA was induced 11-fold by cadmium. While in Zn-supplemented cells exposed to cadmium Hmox1 mRNA levels exceeded controls, the levels were still less than in cells exposed to only cadmium (*p* < 0.01; Zn + Cd vs. Cd). BSO alone did not induce Hmox1 mRNA, and slightly enhanced Hmox1 mRNA induction by cadmium. In Zn-supplemented cells exposed to cadmium with BSO, Hmox1 mRNA levels were similar to those in cells exposed to cadmium alone.

Mt1 mRNA was induced 20-fold by cadmium, whereas in Zn-supplemented cells, cadmium induced Mt1 mRNA 6-fold. BSO alone did not induce Mt1 mRNA, but enhanced cadmium-elicited induction of Mt1 mRNA to levels 60 times greater than control. However, in Zn-supplemented cells, BSO did not enhance Mt-1 induction by cadmium; gene expression was comparable to that in cells exposed to cadmium alone (*p* > 0.45; BSO + Zn + Cd vs. Cd).

As compared to control, Gclc and Gclm mRNA levels were induced 2.6- and 5-fold by cadmium exposure (Figure 6C; *p* < 0.01), but not by Zinc. Supplementation with Zinc attenuated induction of Gclc and Gclm mRNA by cadmium (*p* < 0.02). BSO alone did not induce Gclc and Glcm mRNA, but markedly enhanced Gclm mRNA induction by cadmium (*p* < 0.05; BSO + Cd vs. Cd) without altering Gclc induction. In Zn-supplemented cells, BSO did not enhance Glcm mRNA induction by cadmium; expression was comparable to that in cells exposed to cadmium alone (*p* > 0.37).

Finally, apical choline uptake was compared among CPECs treated with combinations of cadmium, Zinc, and BSO (Figure 7). As compared to control, cadmium increased choline uptake by 62% (*p* < 0.01), and in the presence of BSO, cadmium increased uptake by 106% (*p* < 0.01). In the absence or presence of BSO, Zinc supplementation completely attenuated stimulation of uptake by cadmium, such that uptake was similar to that in control (*p* > 0.45). Neither BSO nor Zinc alone altered choline uptake.

## 3. Discussion

Our prior study in primary cultures of neonatal rat CPECs showed that low, non-lethal cadmium exposure induced cellular stress and increased choline uptake across the apical or CSF membrane [17]. The main findings in this study were as follows. First, at a non-lethal dose, cadmium decreased the GSH:GSSG ratio and elicited an adaptive stress response marked by stress protein induction and up-regulation of GSH synthesis. Second, Zinc supplementation conferred CPECs cytoprotection against cadmium, including attenuation of cadmium-elicited modulation of choline transport. Third, efficacy of Zinc to protect a critical physiological function, such as solute transport, was sustained despite decreased GSH.

In CPECs, exposure to sub-micromolar cadmium elicited a stress response marked by stress protein induction and up-regulation of GSH synthesis and decreased the ratio of reduced GSH to GSSG. This was consistent with induction of oxidative stress [22]. Comparable in vitro exposure of isolated CP tissues to cadmium also elicited induction of stress protein and GCL subunit gene expression. Cadmium indeed can deplete intracellular GSH [13,25]. For example, 25 µM cadmium exposure decreases GSH in HepG2 cells [45]. Nevertheless, exposure to sub-micromolar cadmium increased GSH in rat mesangial cells [46], as did exposures to cadmium as high as 40 µM in rat lung fibroblasts and human H441 bronchiolar cells [47,48]. GSH and GSH:GSSG ratio are assayed typically only after experimental treatment, as done in this study. Analyses of GSH over the course of in vitro and in vivo cadmium exposures indicate GSH may decrease initially but gradually return to or exceed pre-exposure values [49,50,51]. As GSH scavenges ROS, GSH concentration decreases; this alleviates inhibition of GCL and permits de novo replenishment of GSH [22,43]. This may explain our observation in CPECs that were treated with BSO to inhibit GCL and diminish GSH synthesis; in that case, cadmium no longer increased GSH, but actually decreased it. This suggests that cadmium elicited increases in GSH via up-regulation of de novo synthesis in CPECs. Newly synthesized GSH may help manage the oxidative load and directly bind cadmium, thereby dampening the severity of structural or functional injury of the epithelium by cadmium. As shown in HepG2 cells, treatment with α-lipoic acid to regenerate GSH alleviated cadmium cytotoxicity [45].

Cadmium induced both the GCLC and GCLM of GCL, the catalyst of the rate-limiting step and regulatory point in de novo synthesis of GSH [22]. Induction of GCLC gene and protein expression in response to cadmium-elicited oxidative cellular stress has been established [48,52]. Induction of GCLM by metals or cellular stress has not been characterized extensively but can be induced under conditions of oxidative stress [22]. A study in mouse lung demonstrated induction of GCLM by quantum dots with cadmium as the core constituent [53]. GCLM also is induced under conditions of oxidative cellular stress [22]. In mouse hepatoma cells, tert-butylhydroquinone elicited oxidative stress and maximally induced gene expression of GCLM 10-fold, and that of GCLC 2-fold [48]. In cadmium-treated CPECs, regulation of GCL also was predominantly transcriptional, and induction of GCLM was more pronounced than that of GCLC. Thus, in CP, a low-dose cadmium exposure may increase GSH synthesis in part through induction of both GCL subunits.

As is characteristic of cellular stress responses to cadmium [13,54], cadmium elicited incremental induction of HO-1 and Mt-1 and biphasic induction of Hsp70 in CPECs. Cadmium also markedly induced Hmox1, Mt1, and Hspa4 mRNA in isolated neonatal CP tissues. Nonetheless, there was greater induction of Hsp70 mRNA and protein and Mt1 mRNA and enhanced stimulation of choline uptake in cells exposed to cadmium in presence of BSO. This indicated that up-regulation of GCL subunits and GSH synthesis was integral to the adaptive response to cadmium. GSH can directly bind cadmium and lower its effective concentration, but also scavenge free radicals and regulate the oxidative load [23,24,55]. Inhibiting the cell’s ability to replenish GSH might have resulted in greater ROS accumulation, which would accentuate cellular stress and stress protein induction. GCL activity in rat CP is greater than that in various brain regions [26], and despite its small size, CP produces and secretes a significant fraction of GSH in CSF [44]. GSH plays central roles in amino acid transport, drug metabolism, and CSF clearance [26,27]. Furthermore, the GSH antioxidant system in CP appears integral in regulating the ROS load generated at baseline and additional ROS generated by oxidative injury, such as increased H_2_O_2_ in CSF [28]. Our findings support that postulate.

Supplementing CPECs with Zinc conferred cytoprotection against cadmium. Although Zinc did not prevent decreases in the GSH:GSSG ratio elicited by cadmium, the induction of HO-1 and Mt-1 were less pronounced. Furthermore, Hsp70 was not induced, and apical choline uptake was no longer stimulated. Induction of Hsp70 is triggered by protein unfolding and proportional to severity of cellular stress [56,57]. Thus, the decreased Hsp70 induction in Zinc-supplemented cells suggested that Zinc facilitated adaptation to or lessened the severity of cadmium-elicited stress. Prior and sustained Zinc treatment can protect against cadmium-induced cellular injury or dysfunction by various mechanisms [34,47]. In testis interstitial cells, Zinc impaired cadmium-induced carcinogenesis in part by increasing cellular efflux of cadmium [58]. In pulmonary epithelia, Zinc lessened cadmium toxicity by inhibiting cadmium influx via Zinc importer ZIP8 (SLC39A8) [59]. However, similar to our observations in CPECs, others have reported that in H441 bronchial epithelial cells, Zinc supplementation enhanced cadmium resistance without decreasing cellular accumulation of cadmium [47]. Cadmium induces HO-1 transcription via activation and subsequent nuclear export of the transcriptional repressor of HO-1 Bach1 [60]. In CPECs, HO-1 gene induction persisted in cells exposed dually to cadmium and Zinc, indicating Zinc did not broadly disrupt physical or molecular interactions of cadmium within the cell. Although Zinc did not decrease total cadmium accumulation in CPECs, Zinc may minimize effects of cadmium by modulating subcellular cadmium distribution [58].

Preconditioning cells or tissues with non-lethal physicochemical stress sufficient to cause protein unfolding, and thus signal heat shock protein (hsp) induction, may preserve morphological and functional integrity, including the capacity to transport solute against a subsequent stress of comparable or greater severity [61,62,63]. Treating rats with Zinc at doses sufficient to induce Hsp70, Hsp90, and Hsp38 enhanced cadmium tolerance in cultured renal proximal tubules isolated from treated rats [64]. In cultured flounder renal proximal tubule, 6 h of exposure to 100 µM ZnCl_2_ with recovery protected net transepithelial sulfate secretion against subsequent severe heat shock or chemical stress as a consequence of hsp induction [62]. As we previously showed in isolated shark CP, Hsp70 induction was essential to sustain net transepithelial organic anion transport following 6 h of exposure to 50 µM ZnCl_2_ [65]. CP sensitivity to Zinc may vary with species. Here, in rat CPECs, exposure to as much as 50 µM ZnCl_2_ did not stimulate or impair choline uptake, and extended treatments with 25 µM and 10 µM ZnCl_2_ did not induce Hsp70 in rat CPECs. Although induction of Hsp70 per se was not the predominant mechanism of Zinc cytoprotection, induction of Hsp70 was a critical component of the integrative and adaptive cellular stress response in cadmium-exposed CPECs. Indeed, in non-supplemented and Zinc-supplemented cells, when GSH synthesis was inhibited by BSO, there was a compensatory increased induction of the heat shock protein.

Alternatively, Zinc can facilitate cytoprotection by regulating GSH metabolism [34,36]. In brains of fetal rats of marginally Zinc-deficient dams and IMR-32 neuroblastoma cells maintained in Zinc-deficient media, GCLC and GCLM gene and protein expression and GSH levels were decreased. Zinc-deficient culture conditions also exacerbated susceptibility to dopamine-induced oxidative stress and impeded up-regulation of GSH synthesis, likely due to impaired NRF2 activation [34,37]. Conversely, Zinc supplementation can bolster GSH metabolism. Treating rats with cadmium and Zinc increased liver GSH levels and ratios of GSH to GSSG and decreased liver GSSG levels, as compared to treatment with only cadmium [66]. In retinal pigment epithelial cells, 5 µM ZnCl_2_ increased GCLC and GLCM gene expression and GSH levels [67]. However, in CPECs, micromolar ZnCl_2_ did not increase GSH synthesis, induce GCL subunits, or disrupt up-regulation of GSH chemistry by cadmium. Furthermore, in Zinc-supplemented CPECs in which GSH synthesis was impaired by BSO, the induction of Hsp70, HO-1, and Mt-1 by cadmium was no longer accentuated, and stimulation of choline uptake was still diminished. Thus, Zinc can facilitate adaptation to cellular stress to a low-dose cadmium exposure in a manner independent of GSH status.

Zinc supplementation markedly induced metallothionein (Mt1) mRNA expression. Though not elucidated here, increased metallothionein (MT) protein expression before cadmium exposure is a possible mechanism by which Zinc facilitated stress adaptation and abated modulation of apical choline uptake in CPECs. In primary mouse hepatocytes, Zinc pretreatment to induce MT expression before chemical induction of oxidative stress with ferric nitriloacetate minimized cellular stress despite GSH depletion with BSO [68]. This method has also been shown to curtail cellular injury by cadmium and other agents, such as acetaminophen, in the liver [68,69,70]. MT directly binds Zinc, and in coordination with Zinc transporters, regulates free cytosolic Zinc and its distribution among specific sub-cellular compartments and Zinc-dependent enzymes [34,39,71]. Supplementation increases intracellular Zinc and signals synthesis of new MT that can bind additional Zinc. MT binds up to seven Zinc ions, but in normal redox balance not all metal-binding sites are occupied. The protein has even greater affinity for cadmium. Thus, if MT were up-regulated before cadmium exposure, it could bind cadmium and decrease its effective concentration [55]. MT also scavenges and thereby regulates accumulation of ROS produced as cadmium inhibits electron transport and displaces Fenton metals [72]. Although Zinc is redox inert, Zinc–thiol binding within MT is redox-sensitive. Under oxidizing conditions such as that induced by cadmium, Zinc may be released and then function in varying capacities to facilitate redox signaling and antioxidant defenses [39]. A more targeted investigation of MT in Zinc-mediated adaptation to cellular stress elicited by cadmium in CP is warranted, but is beyond the scope of the current study.

## 4. Materials and Methods

### 4.1. Animal Use and Tissue Harvest

With approval from the Institutional Animal Care and Use Committee at Texas A&M University (protocol no. 2011-128), choroid plexus (CP) tissues were harvested from 2–3-day-old neonatal Sprague–Dawley rats obtained from time-pregnant dams (Charles River, Roanoke, IL, USA). On gestational day 16, dams were acquired and maintained in a 12 h light/dark cycle with free access to regular chow and water. For each primary culture preparation, lateral and fourth CP tissues were harvested from 36–50 neonatal rat brains with sterile instruments and pooled and initially maintained in chilled collection medium. For in vitro experiments in isolated intact CP, tissues from neonatal rats also were pooled and maintained initially in chilled collection medium.

### 4.2. Chemicals, Reagents, and Solutions

All chemicals were analytical grade and purchased from commercial vendors. Critical reagents and kits were purchased from the following vendors: CdCl_2_ and ZnCl_2_, Sigma (St. Louis, MO, USA); *l*-buthionine sulfoximine, Acros Organics (Morris Plains, NJ, USA); and GSH/GSSG-Glo™ Assay (Promega, Madison, WI, USA). The collection medium consisted of sterile DMEM/F12 medium with 100 units penicillin per 100 mL. The dissociation buffer contained (in mM) 137 NaCl, 2.7 KCl, 0.7 Na_2_HPO_4_, 5.6 glucose, 10 HEPES (pH 7.4), 5 U/mL protease (Sigma, St. Louis, MO, USA), and 1500 kU/mL DNase I (Calbiochem-EMD Millipore, Billerica, MA, USA). The plating medium consisted of minimum essential medium with d-valine substituted for l-valine (U.S. Biological, Swampscott, MA, USA), 10% NuSerum IV (BD Biosciences, San José, CA, USA), 100 ng/mL PGE1, 10 µM forskolin, 1.5 µM triiodothyronine, and 50 ng/mL EGF; growth factors were tissue culture grade and purchased from Sigma (St. Louis, MO, USA). The maintenance medium consisted of DMEM-F12 with 5% NuSerum IV and listed growth factors at the same concentrations. Plating and maintenance media were antibiotic- and fungicide-free. Stock solutions of ~1.5 mM CdCl_2_ in sterile ultra-pure water were prepared bi-weekly and diluted serially to a final concentration of 500 nM CdCl_2_ in sterile serum-free DMEM/F12 to prepare treatment media the day cadmium exposure was initiated. Stock solutions of ~3 mM ZnCl_2_ and stock solutions of ~75 mM BSO in sterile ultrapure water were prepared weekly and used directly to make treatment media. The radiolabeled choline transport was assayed in artificial cerebrospinal fluid (aCSF) containing (in mM) 137.4 sodium, 3 potassium, 1.4 calcium, 0.8 magnesium, 0.7 phosphate, 125.4 chloride, 2 urea, 18 bicarbonate, and 10 TRIS/HEPES (pH 7.4) with 12 mM glucose, 10 µM unlabeled choline chloride (Tokyo Chemical Industry, Tokyo, Japan), and trace [^3^H]choline chloride (0.075 µCi, ~80 Ci/mmol; Perkin-Elmer, Waltham, MA, USA). For protein analysis, the cell lysis buffer contained 50 mM TRIS-HCl pH 6.8, 100 mM dithiothreitol, 30% *v*/*v* glycerol, 2% *w*/*v* SDS, 0.05% *v*/*v* Triton X-100, and 0.5% *w*/*v* bromophenol blue; the stripping solution contained 25 mM glycine-HCl, 1% *w*/*v* SDS (pH 2). Hank’s Balanced Salt Solution (HBSS), phosphate-buffered saline (PBS), and TRIS-buffered saline (TBS) of standard compositions were used for various assays.

### 4.3. Isolation and Primary Culture of Choroid Plexus Epithelial Cells (CPECs)

Epithelial cells were isolated from neonatal rat CP tissues by enzymatic and mechanical disruption and cultured by an aseptic technique, as described previously [17,73,74]. Briefly, tissues suspended in dissociation buffer were shaken (37 °C water bath) and triturated intermittently over a 20 min period. The mixture of released cells and extraneous tissue was filtered through a sterile 100 µM mesh cell strainer (BD Biosciences, San José, CA, USA). The filtrate was centrifuged, and pelleted cells were washed with penicillin-supplemented DMEM/F12. Cells were suspended in penicillin-supplemented DMEM/F12 with 10% NuSerum IV and pre-plated in a 35 mm Petri dish for 3.5 h at 37 °C (humidified 95% air/5% CO_2_); during this time extraneous fibroblasts attached, thereby minimizing fibroblast contamination of the CP epithelial cell culture. Unattached epithelial cells then were suspended in plating medium, plated on impermeable supports; i.e., non-coated sterile polystyrene tissue culture plates (Corning, Suwanee, GA, USA), at 3 × 10^5^ cells/cm^2^ in 300 µL medium/cm^2^ plating surface, and incubated (37 °C; humidified 95% air/5% CO_2_). The base of plating medium was MEM with d-valine substituted for l-valine; d-valine is poorly metabolized by fibroblasts, which limits fibroblast survival and proliferation; this further minimized fibroblast contamination of CP epithelial primary cultures [75]. After 72 h, plating medium and unattached cells were removed and replaced with maintenance medium; thereafter, medium was replaced every two days until start of experimental treatment. CPECs formed fully differentiated confluent monolayers within 6 days, and experiments were initiated 6–9 days post-plating. Cells remained in maintenance medium until initiation of experimental treatments.

### 4.4. Treatment of CPECs and Isolated CP Tissues with Cadmium, Zinc, and BSO

CPECs were maintained in complete maintenance medium until initiation of experimental treatment. To study the effects of cadmium in CPECs, cells were exposed to 500 nM CdCl_2_ for 12 h. We selected this concentration and duration of cadmium exposure based our prior investigation of time-dependent and concentration-dependent exposure of CPECs to 0–1000 nM CdCl_2_ [17]. Twelve-hour exposure to 500 nM CdCl_2_ induced a cellular stress response and marked stimulation in apical choline uptake, but did not increase lactate dehydrogenase release; i.e., caused minimal cytotoxicity. The protocol for Zinc treatment was based on a series of experiments that determined a non-toxic dose of Zinc after 48 and 72 h of incubation.

Experimental treatments of CPECs and isolated CP tissues implemented in this study are described here in detail and are outlined in a schematic diagram in Figure 7. All treatments were performed at 37 °C (humidified 95% air/5% CO_2_). The potential for Zinc supplementation to modify cadmium-induced responses in CPECs was examined under conditions in which Zinc was present both before and during cadmium exposure. As shown in Figure 1A, CPECs were first incubated for 48 h in maintenance medium without Zinc supplement; these cells were designated as non-supplemented cells. In parallel, another group of CPECs was incubated for 48 h in maintenance medium with 25 µM ZnCl_2_; these cells were designated as Zn-supplemented cells. Non-supplemented cells were then pre-treated in Zinc-free SFM for 12 h and divided into two sets: one set was incubated for 12 h in fresh Zinc-free SFM with 0 CdCl_2_ (Control), while the other set was exposed for 12 h to 500 nM CdCl_2_ in fresh Zinc-free SFM (Cadmium, Cd). In parallel, Zn-supplemented cells were pre-treated in SFM with 10 µM ZnCl_2_ and then divided into two sets: one set was incubated for 12 h in fresh SFM with 10 µM ZnCl_2_ (Zinc, Zn), while the other was exposed for 12 h to 500 nM CdCl_2_ in SFM with 10 ZnCl_2_ (Zn + Cd) (Figure 1A).

Time-dependent induction of GCLC, GCLM, and stress protein mRNA and protein expression were analyzed in non-Zinc supplemented CPECs exposed to cadmium. CPECs were first incubated for 48 h in complete maintenance medium without Zinc supplement; i.e., non-supplemented cells. These cells were subsequently pre-treated for 12 h in SFM and then exposed to 0 (*Control*) or 500 nM CdCl_2_ (Cd) in SFM for up to 12 h (Figure 1B). At 3, 6, 9, and 12 h, representative cadmium-exposed and time-matched control cells were collected for analysis of mRNA and protein expression by quantitative real-time polymerase chain reaction or immunoblot analysis, respectively.

To investigate in vitro effects of cadmium in intact CP, lateral and fourth CP tissues from four neonatal rats were harvested and pooled; this comprised a single sample (i.e., *n* = 1). Each set of tissues was collected initially in penicillin-supplemented DMEM-F12 without serum and then rinsed generously with sterile penicillin-supplemented PBS (600 mg/100 mL). Tissues were transferred onto a pre-wetted polycarbonate permeable cell support (Thermo Scientific-Nunc, Waltham, MA, USA) in a well of a sterile 6-well plate with 4 mL SFM to equilibrate (2 h, 37 °C; humidified 95% air/5% CO_2_). Equilibration medium was replaced with 4 mL SFM with 0 or 500 nM CdCl_2_; tissues were incubated for 24 h (37 °C; humidified 95% air/5% CO_2_) (Figure 1C).

Time-dependent effects of Zinc on apical [^3^H]choline uptake were examined in CPECs that were first incubated for 48 h in maintenance medium not supplemented with ZnCl_2_. Representative cells were subsequently treated for 24 h in SFM containing 0 (Control), 5, 10, 25, 50, or 100 µM ZnCl_2_ (Figure 1D).

To characterize the role of GSH in cellular responses to cadmium and Zinc supplementation, non-Zinc-supplemented and Zinc-supplemented cells were treated with the *gamma*-cysteine ligase inhibitor, *l*-buthionine sulfoximine (BSO) (Figure 1E). We used 100 µM BSO in our prior study of the effects of cadmium in CPECs [17]. Studies in T-lymphocytes had indicated BSO was effective in the range of 10–100 µM [76]. CPECs were first incubated for 48 h in maintenance medium without Zinc; i.e., non-supplemented cells, or with 25 µM ZnCl_2_; i.e., Zn-supplemented cells. To test the effects of cadmium and Zinc supplementation under conditions in which GSH synthesis was inhibited before and during cadmium exposure, 100 µM BSO was added to the SFM used for pre-treatment and cadmium exposure of non-supplemented and Zn-supplemented CPECs. In the absence of BSO, a group of non-supplemented cells was pre-treated (12 h) in SFM and then divided into two sets; one was incubated for 12 h in SFM without cadmium (Control), while the other was exposed for 12 h to 500 nM CdCl_2_ in SFM (Cd). Also in absence of BSO, a group of Zn-supplemented cells was pre-treated (12h) in SFM with 10 µM ZnCl_2_ and then divided into two sets: one was incubated for 12 h with 10 µM ZnCl_2_ alone (Zn), while the other was exposed for 12 h to 500 nM CdCl_2_ in SFM with 10 µM ZnCl_2_ (Zn + Cd). In the presence of BSO, another group of non-supplemented cells was pre-treated (12 h) in SFM with 100 µM BSO and then divided into two sets; one set was incubated for 12 h in SFM without cadmium (BSO), while the other was exposed for 12 h to 500 nM CdCl_2_ in SFM with 100 µM BSO (BSO + Cd). In the presence of BSO as well, a group of Zn-supplemented cells was pre-treated (12 h) in SFM with 100 µM BSO and 10 µM ZnCl_2_ and then divided into two sets; one set was exposed (12 h) in SFM with 100 µM BSO and 10 µM ZnCl_2_ and no cadmium (BSO + Zn), while the other was exposed (12 h) to 500 nM CdCl_2_ in SFM with 100 µM BSO and 10 µM ZnCl_2_ (BSO + Zn +Cd). All treatments were performed at 37 °C. After treatment, cells were processed for analysis of glutathione chemistry, mRNA expression, and apical uptake of [^3^H]choline.

### 4.5. Analysis of Cellular Accumulation of Zinc and Cadmium

Cells grown in 12-well plates were treated with 1 mL of experimental media. Total accumulation of elemental Zinc and elemental cadmium was determined by inductively coupled plasma-mass spectrometry (ICP-MS). The protocol for elemental metal analysis in cultured cells by ICP-MS is the standard analysis protocol used in the Trace Element Research Laboratory (College of Veterinary Medicine & Biomedical Sciences, Texas A&M University–College Station), which performed the analysis. This same protocol was used to determine cellular cadmium content in CPECs in our prior investigation of cadmium-induced stress responses in CPECs [17]. Treated cells were rinsed twice with chilled PBS containing 1 mM EDTA and once with chilled PBS (no EDTA). Within the original culture plate, cells were solubilized in 200 µL concentrated nitric acid (Ultrex grade). The entire cell suspension was aspirated and transferred to a pre-weighed 15 mL tube (±0.1 mg); each well was rinsed with ultra-pure water several times, and each rinse was transferred to the 15 mL tube. Water was added to the tube to a final volume of 10 mL and final nitric acid concentration of 2%, and the final weight was recorded (±0.1 mg). Samples were analyzed by ICP-MS (ELAN DRC-II inductive coupled plasma mass spectrometer, PerkinElmer Life and Analytical Sciences, Shelton, CT, USA) in the Trace Element Research Laboratory (College of Veterinary Medicine & Biomedical Sciences, Texas A&M University–College Station); mixed reference standards of Zinc and cadmium in 2% HNO_3_ were analyzed to calibrate the instrument. Representative cells from the control and experimental groups were collected to determine protein content by Bradford assay with BSA standards (Bio-Rad, Hercules, CA, USA). Total cellular accumulation of each element was expressed as nanograms per milligram protein and analyzed in duplicate in three separate culture preparations (*n =* 3).

### 4.6. Intracellular Reduced Glutathione and Oxidized Glutathione Assay

Total intracellular reduced glutathione (GSH) and oxidized glutathione (GSSG) concentrations were measured by the GSH/GSSG-Glo™ Assay (Promega, Madison, WI, USA), as per the protocol for adherent cells. CPECs were grown in flat-bottomed, white-walled 96-well plates (Corning, Suwanee, GA, USA). After treatment, cells were rinsed with HBSS and lysed in 50 µL Total Glutathione Reagent or Oxidized Glutathione Reagent with vigorous rocking (5 min; RT). Then, 50 µL Luciferin Generation Reagent was added to each well, and the plate was incubated (30 min, RT). Finally, 100 µL Luciferin Detection Reagent was added to each well, and the plate was incubated (45 min, RT) before determination of luminescence (Tecan Infinite M200 microplate reader, Morrisville, NC, USA). To quantify intracellular GSH and GSSG concentrations and calculate the ratio of GSH to GSSG (*GSH:GSSG*), relative light units (RLUs) of each sample were compared to RLUs determined for a set of GSH standards (0, 0.25, 0.50, 1, 2, 4, 8, and 16 µM). GSSG concentration was determined by extrapolated calculation based on the GSH standard curve.

### 4.7. Analysis of Gene Expression

Gene expression was analyzed by quantitative real-time polymerase chain reaction (qRT-PCR) with SYBR^®^ Green detection, as described previously [17]. Total RNA was extracted from CPECs and isolated CP tissues with a RNeasy mini kit (Qiagen, Valencia, CA, USA). CPECs grown and treated in 12-well plates were rinsed with PBS and triturated in 500 µL RLT lysis buffer/1% β-mercaptoethanol (β-ME); CP tissues were rinsed with PBS and triturated in 600 µL RLT/1% β-ME. Cell and tissue lysates were homogenized by centrifugation through a QiaShredder. Homogenate was cleared with 70% EtOH and loaded onto a RNeasy Spin column for DNase treatment (Qiagen, Valencia, CA, USA) and final elution of total RNA with RNase-free water. Each RNA sample was evaluated for quality and contamination per nanospectroscopic determination of Abs260 nm:Abs 280 nm and Abs260 nm:Abs230 nm ratios (NanoDrop^®^ ND-100, Thermo Scientific, Waltham, MA, USA) before preparation of first-strand cDNA (iScript cDNA Synthesis kit; Bio-Rad, Hercules, CA, USA). In each cDNA sample, each test gene was analyzed in triplicate in 4–5 separate culture preparations by qRT-PCR with SYBR^®^ Green detection (MyQ single-color real-time detection system, Bio-Rad, Hercules, CA, USA). Copy number for each gene was determined and normalized to integrated values for Actb (β-actin) and Gapdh (GAPDH) mRNA expression; expression of these genes remained relatively stable and consistent under control and experimental conditions in our studies. Specifications for qRT-PCR analysis: initial denaturation—10 min, 95 °C; amplification and quantification (45 cycles)—15 s, 95 °C; 30 s, 60 °C; melt curve—55 °C–95 °C. Primers were designed by Primer Express software (PE Applied Biosystems, Foster City, CA, USA) based on GenBank sequences for rat genes Actb, Gapdh, Hspa4, Hmox1, Mt1, Gclc, and Gclm, synthesized by Integrated DNA Technologies (Coralville, IA, USA), and tested using cDNA from untreated cultured cells to ensure primer–primer dimerization did not occur. Forward and reverse primer sequences are listed in Table 5.

### 4.8. Immunoblot Analysis

Protein expression in CPECs grown and treated in 48-well or 96-well plates was determined by immunoblot analysis, as described previously [17]. Cells were solubilized in chilled lysis buffer with phosphatase and protease inhibitor cocktail, heat-denatured, sonicated, and centrifuged. Cellular proteins and molecular weight markers were electrophoresed (10% SDS-PAGE) and electroblotted onto a polyvinylidene fluoride membrane by tank transfer. Membranes were blocked (1 h, RT) with 5% or 10% non-fat dry milk (NFDM) in TBS with 0.1% Tween 20 (TBS-T), then incubated in 5% or 10% NFDM/TBS-T with primary antibody (2 h, RT or 18 h, 4 °C). Membranes were rinsed in TBS-T and incubated with goat-derived secondary antibody conjugated with alkaline-phosphatase (AP) or horseradish peroxidase (HRP; 90 min, RT). Immunoreactivity was detected by chromogenic staining with AP substrate 5-bromo-4-chloro-3-indolyl-phosphate/nitro blue tetrazolium (BCIP/NBT; Promega, Madison, WI, USA) or enhanced chemiluminescence with HRP substrate (Millipore, Bedford, MA, USA). Protein bands were analyzed by densitometry (Alpha Innotech FluoroChem HD2 gel documentation system, Santa Clara, CA, USA). Membranes were cleared of primary and secondary antibodies and probed for additional proteins. Primary antibodies against β-actin (Cat. No. A4700, Sigma-Aldrich, St. Louis, MO, USA), HO-1 (Cat. No. ADI-OSA-150, Enzo Life Sciences Farmingdale, NY, USA), and Hsp70 (Cat. No. ADI-SPA-812, Enzo Life Sciences, Farmingdale, NY, USA) were used at 1:1000 (10% NFDM/TBS-T). Primary antibodies against GCLC (Cat. No. ab41463, Abcam, Cambridge, MA, USA) and GCLM (Cat. No. ab126704, Abcam, Cambridge, MA, USA) were used at 1:500 (5% NFDM/TBS-T). AP-conjugated anti-mouse IgG secondary antibody (Cat. No. AP124A, Millipore-Sigma, Burlington, MA, USA) or HRP-conjugated anti-rabbit IgG secondary antibody (Cat. No. ADI-SAB-300, Enzo Life Sciences, Farmingdale, NY, USA) were used at 1:5000 in 10% or 5% NFDM/TBS-T.

### 4.9. Radiotracer Assay of Apical Choline Transport in CPECs

Thirty-minute apical uptake of [^3^H]choline was assayed in cells plated in 48-well tissue plates, as described previously [17]. After treatment, cells were rinsed with aCSF and incubated (30 min, 37 °C, 95% air/5% CO_2_) with 200 µL aCSF containing 10 µM unlabeled choline chloride and trace [^3^H]choline chloride ± 750 µM hemicholinium-3 (HC-3); the uptake buffer did not contain experimental agents; i.e., cadmium, Zinc, or BSO. Uptake was terminated by removal of buffer and a triple rinse with ice-cold isotope-free aCSF containing 5 mM choline chloride; this also removed residual isotope from the culture well. Cells were solubilized in 200 µL 1 M NaOH and neutralized with 200 µL 1N HCl. Two 50 µL aliquots of solubilized cell suspension were collected to determine radioactivity by liquid scintillation (Beckman LC 6500 Scintillation Counter, Fullerton, CA, USA). One 50 µL aliquot of cell suspension was collected to determine protein content (Bradford assay with BSA standard, Bio-Rad, Hercules, CA, USA). [^3^H]choline uptake was calculated as picomoles [^3^H]choline per milligram protein. Uptake was measured in triplicate (triplicate measures) in at least three separate culture preparations (*n* = 3). Data are reported as a percentage of mediated choline uptake by control cells; means ± SE. We had previously determined that in CPECs 24 h serum-deprivation does not alter 30 min apical uptake of 10 µM [^3^H]choline, as compared to that in cells incubated with 5% NuSerum [17].

### 4.10. Extracellular Lactate Dehydrogenase Assay

Cytotoxicity was evaluated based on increased release of lactate dehydrogenase (LDH) from experimental versus control cells using the CytoTox 96^®^ Non-Radioactive Cytotoxicity Assay (Promega, Madison WI, USA). Cells grown in 48-well plates were incubated with 400 µL treatment medium. Maximum LDH release was determined in non-treated control cells lysed with 0.9% *v*/*v* Triton X-100 (45 min, 37 °C). After treatment, three 50 µL aliquots of treatment medium from each control and experimental well were transferred to individual wells of a 96-well plate. Substrate solution (50 µL) was added to each sample; the mixture was incubated in the dark (10 min, RT). Stop solution (50 µL) was added to each well, and Abs490 nm was recorded at 24 °C (Tecan Infinite M200 microplate reader, Morrisville, NC). Values were corrected for background absorbance; i.e., cell-free, serum-free DMEM/F12. LDH release was expressed as a percentage of maximal LDH release; LDH release was assayed in triplicate in three separate culture preparations (triplicate measures; *n* = 3).

We assessed cytotoxicity of cadmium exposure and Zinc supplementation in presence of BSO by colorimetric assay of extracellular LDH release (Table 6). In control cells mean LDH release was 11% of maximal release. Cadmium insignificantly increased LDH release to 14%. This was consistent with findings in our previous study [17]. Zinc did not alter LDH release, either alone or in cadmium-exposed cells. Similarly, BSO did not alter LDH release or alter release in cadmium-exposed cells. These data indicated treatment with these agents, individually or in combination, did not elicit marked cytotoxicity.

### 4.11. Statistical Analysis

Results are reported as means ± SE of 3–5 experiments, each conducted in separate culture preparations or separate sets of isolated choroid plexus tissues. Statistical analyses were performed using JMP^®^ statistical software by SAS (Cary, NC, USA). Control and experimental means were compared by one-way or two-way analysis of variance (ANOVA) or Student’s *t*-test. Control and experimental means from experiments that evaluated effects of cadmium and Zinc treatments on metal accumulation, GSH chemistry, and apical choline uptake were compared by one-way ANOVA with a Tukey–Kramer post hoc test. Control and experimental means from experiments that evaluated time-dependent effects of cadmium on mRNA expression and protein expression in CPECs were compared by two-way ANOVA with a Tukey–Kramer post hoc test. If there was a significant group by time interaction, then group differences at each time point were compared using Student’s *t*-test. Control and experimental means from experiments that evaluated concentration-dependent effects of Zinc on apical choline uptake were compared by one-way ANOVA with a Tukey–Kramer post hoc test. Control and experimental means from experiments that compared mRNA expression for each gene of interest in control and cadmium-treated isolated choroid plexus tissues were compared by a two-tailed Student’s *t*-test. Similarly, control and experimental means from experiments that compared mRNA expression for each gene of interest in non-supplemented versus Zinc-supplemented cells that were not subjected to additional treatment with cadmium or BSO were compared by a two-tailed Student’s *t*-test. Control and experimental means from experiments that investigated the effects of cadmium with manipulation of glutathione synthesis and Zinc supplementation were compared by one-way ANOVA with a Tukey–Kramer post hoc test. Differences were deemed significant at *p* < 0.05.

## 5. Conclusions

In summary, a low-dose cadmium exposure sufficient to induce cellular stress stimulated choline uptake across the apical membrane of CPECs. Cells mounted an adaptive cellular stress response defined by a progressive induction of heme oxygenase protein and gene expression and metallothionein gene expression, along with biphasic induction of Hsp70 integrated with concurrent up-regulation of GSH synthesis. Supplementation with Zinc did not prevent onset of cellular stress elicited by cadmium, but facilitated adaptation and diminished stimulation of choline uptake independent of GSH. This study lends insight into how neonatal and possibly adult CP epithelium may sequester contaminant heavy metals, but not succumb to cellular injury and modulation by the very agents from which it protects the brain. Our data indicated that Zinc did not impair the capacity of CPECs to accumulate cadmium. Thus, Zinc is not only protective of choroid plexus biology, but could potentially protect this epithelium, as it still sequesters toxic metals and continues to protect the brain from their neurotoxic effects. These data also suggest an animal’s Zinc status and capacity of CP epithelium to accumulate Zinc may dictate the efficacy of cytoprotective mechanisms that minimize cellular injury and dysfunction elicited by cellular stress.

## Figures and Tables

**Figure 1 ijms-22-08857-f001:**
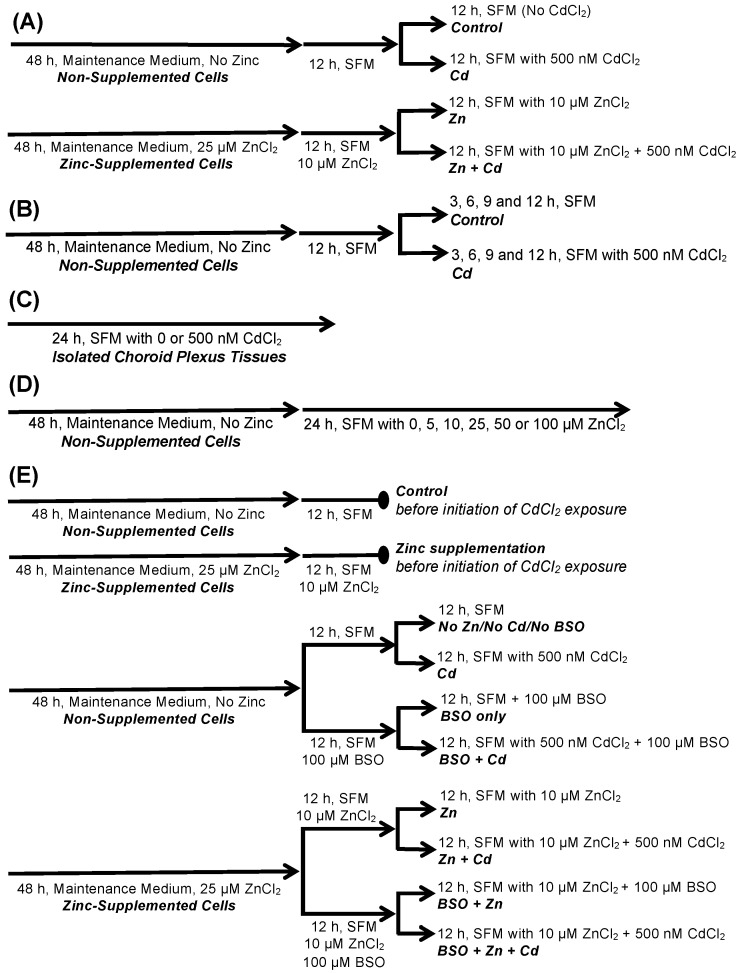
Schematic outlines for experimental treatments of choroid plexus epithelial cells (CPECs) or isolated choroid plexus (CP) tissues with cadmium (Cd), Zinc (Zn), and *l*-buthionine sulfoximine (BSO). (**A**) CPECs were incubated (48 h) in maintenance medium with 0 or 25 µM ZnCl_2_ to prepare non-supplemented versus Zn-supplemented cells, respectively. Non-supplemented cells were then pretreated (12 h) in serum-free medium (SFM) before 12 h exposure in SFM to either 0 or 500 nM CdCl_2_; Zn-supplemented cells were pretreated (12 h) in SFM with 10 µM ZnCl_2_ before 12 h exposure in SFM with 10 µM ZnCl_2_ and either 0 or 500 nM CdCl_2_. (**B**) Non-supplemented CPECs were prepared and pre-treated (12 h, SFM) before 12 h exposure in SFM with either 0 or 500 nM CdCl_2_; cadmium and time-matched control cells were collected at 3, 6, 9, and 12 h. (**C**) CP tissues from neonatal rats were isolated and incubated (24 h) in SFM with 0 or 500 nM CdCl_2_. (**D**) Non-supplemented CPECs were prepared and then incubated (24 h, SFM) with 0-100 µM ZnCl_2_. (**E**) Non-supplemented and Zn-supplemented CPECs were prepared and exposed to cadmium in the absence or presence of BSO, an inhibitor of glutathione synthesis. Representative non-supplemented and Zn-supplemented cells were treated (12 h) in SFM without or with 10 µM ZnCl_2_ respectively, but not subjected to further treatment. Other cells were treated as follows. In absence of BSO, a group of non-supplemented cells was pre-treated (12 h, SFM) and then exposed (12 h) in SFM to either 0 or 500 nM CdCl_2_. In absence of BSO as well, a group of Zn-supplemented cells was pre-treated (12 h, SFM) with 10 µM ZnCl_2_ and then exposed (12 h) in SFM with 10 µM ZnCl_2_ to either 0 or 500 nM CdCl_2_. In the presence of BSO, another group of non-supplemented cells was pre-treated (12 h) in SFM with 100 µM BSO and then exposed (12 h) in SFM with 100 µM BSO to either 0 or 500 nM CdCl_2_. In the presence of BSO as well, a group of Zn-supplemented cells was pre-treated (12 h) in SFM with 100 µM BSO and 10 µM ZnCl_2_ and then exposed (12 h) in SFM with 100 µM BSO and 10 µM ZnCl_2_ with either 0 or 500 nM CdCl_2_. All treatments were performed at 37 °C.

**Figure 2 ijms-22-08857-f002:**
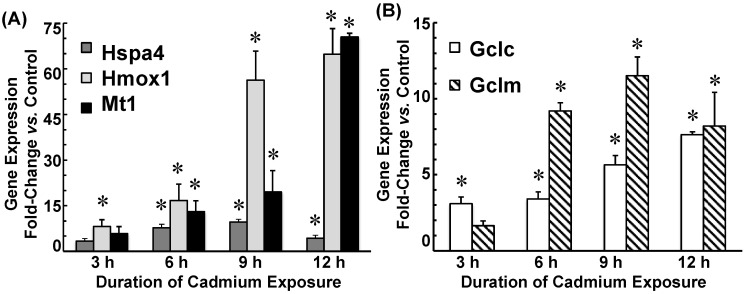
Time-dependent gene expression for rat heat shock protein-70 (Hspa4), heme oxygenase-1 (Hmox1), and metallothionein (Mt1) (**A**) and for GCLC (Gclc) and GCLM (Gclm) (**B**) in choroid plexus epithelial cells (CPECs) exposed to 500 nM CdCl_2_. Non-Zinc supplemented CPECs were pre-treated (12 h) in serum-free medium (SFM) and then divided into two sets: one set was incubated for 12 h in SFM without CdCl_2_ (Control), while the other was exposed for 12 h to 500 nM CdCl_2_ in SFM (Cd). All treatments were performed at 37 °C. At 3, 6, 9, and 12 h, representative cadmium-exposed and time-matched control cells were collected for analysis of mRNA expression by quantitative real-time polymerase chain reaction (qRT-PCR); mRNA expression for each test gene was normalized to mRNA for rat β-actin (Actb) and GAPDH (Gapdh). Fold-induction was calculated as the ratio of normalized gene expression in Cd-exposed cells to that in time-matched controls. mRNA was analyzed in triplicate in five separate culture preparations. Data are means ± SE; *n* = 5; * *p* < 0.05 vs. time-matched Control.

**Figure 3 ijms-22-08857-f003:**
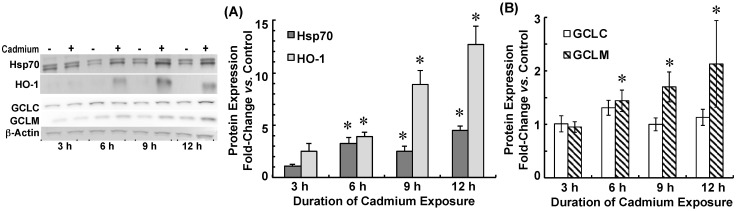
Time-dependent heat shock protein-70 (Hsp70) and heme oxygenase (HO-1) protein expression (**A**) and GCLC and GCLM protein expression (**B**) in choroid plexus epithelial cells exposed to 500 nM CdCl_2_. Non-Zinc supplemented cells were pre-treated (12 h) in serum-free medium (SFM) and divided into two sets: one set was incubated in SFM without CdCl_2_, while the other was exposed to 500 nM CdCl_2_ in SFM (Cd). All treatments were performed at 37 °C. At 3, 6, 9, and 12 h, representative cadmium-exposed and time-matched control cells were collected for immunoblot analysis of proteins of interest and β-actin. A representative immunoblot is shown. Immunoreactivity of each protein with the respective primary antibody was visualized by colorimetric or enhanced chemiluminescence detection. Protein band intensity was normalized to that of β-actin; induction was expressed as fold-change versus time-matched control. Protein expression was analyzed in five separate culture preparations; data are means ± SE; *n* = 5. * *p* < 0.05 vs. time-matched Control.

**Figure 4 ijms-22-08857-f004:**
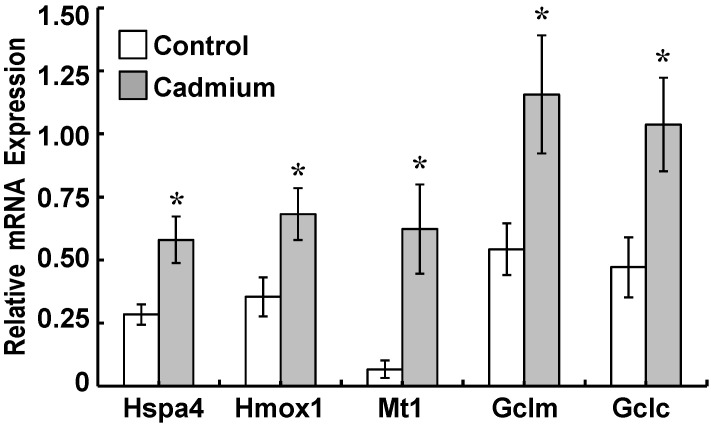
Gene expression for rat heat shock protein-70 (Hspa4), heme oxygenase (Hmox1), metallothionein (Mt1), GCLM (Gclm), and GCLC (Gclc) in isolated neonatal rat lateral and fourth choroid plexus (CP) tissues exposed in vitro to cadmium. Lateral and fourth CP tissues from four neonatal rats were harvested and pooled; this comprised a single sample (i.e., *n* = 1). Each set of tissues was incubated in serum-free medium (SFM) with 0 or 500 nM CdCl_2_ for 24 h (37 °C). The mRNA was analyzed in triplicate by qualitative RT-PCR in six separate sets of CP tissues. The mRNA expression for each test gene was normalized to mRNA for rat β-actin (Actb) and GAPDH (Gapdh). Values are expressed as relative mRNA expression for control and cadmium-treated tissues. Data are means ± SE; *n* = 6 sets of tissues. * *p* < 0.05 vs. Control.

**Figure 5 ijms-22-08857-f005:**
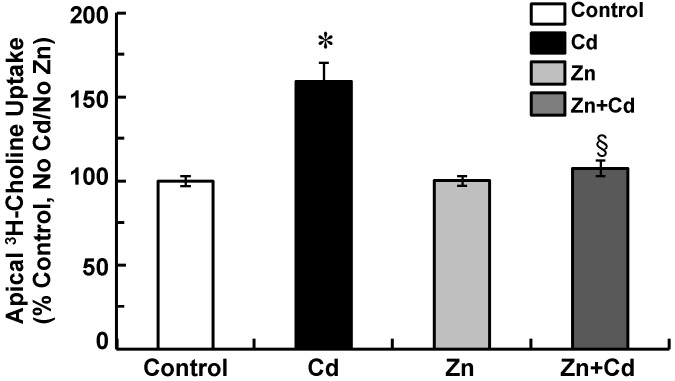
Thirty-minute apical uptake of 10 µM [^3^H]choline in Zinc-supplemented choroid plexus epithelial cells (CPECs) exposed for 12 h to 500 nM CdCl_2_. CPECs were first incubated for 48 h in maintenance medium without Zinc, i.e., non-supplemented cells, or with 25 µM ZnCl_2_, i.e., Zn-supplemented cells. Non-supplemented cells were then pre-treated (12 h) in SFM and divided into two sets: one was incubated for 12 h in SFM with 0 CdCl_2_ (Control), while the other was exposed for 12 h to 500 nM CdCl_2_ in SFM (Cd). In parallel, Zn-supplemented cells were pre-treated (12 h) in SFM with 10 µM ZnCl_2_ and divided into two sets: one set was exposed for 12 h in SFM with 10 µM ZnCl_2_ (Zn), while the other was exposed for 12 h to 500 nM CdCl_2_ in SFM with 10 µM ZnCl_2_ (Zn + Cd). All treatments were at 37 °C. After experimental treatments, cells were rinsed and incubated (30 min, 37 °C) in artificial CSF (10 mM Tris-HEPES, pH 7.4) with 10 µM [^3^H]choline chloride ± 750 µM hemicholinium-3. Uptake was measured in triplicate in three different culture preparations (*n* = 3). Uptake was expressed as percentage of control; data are means ± SE; *n* = 3. * *p* < 0.05 vs. Control. § *p* < 0.05 vs. Cd.

**Figure 6 ijms-22-08857-f006:**
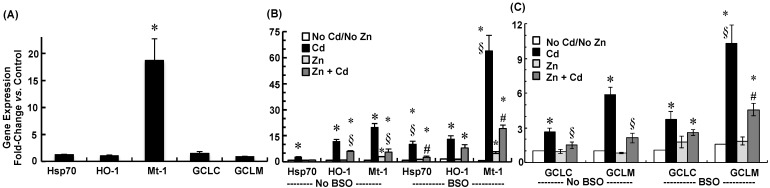
Effects of Zinc supplementation and inhibition of glutathione synthesis by *l*-buthionine sulfoximine (BSO) on gene expression for rat heat shock protein-70 (Hspa4), heme oxygenase (Hmox1), metallothionein (Mt1), GCLC (Gclc), and GCLM (Gclm) expression in choroid plexus epithelial cells (CPECs) exposed for 12 h to 500 nM CdCl_2_. CPECs were first incubated (48 h) in maintenance medium without Zinc; i.e., non-supplemented cells, or with 25 µM ZnCl_2_; i.e., Zn-supplemented cells. (**A**) A representative group of non-supplemented CPECs was treated in serum-free medium (SFM) for 12 h, but not subjected to further treatment; in parallel, representative Zn-supplemented cells were treated in SFM with 10 µM ZnCl_2_ for 12 h, but not subjected to further treatment. (**B**,**C**) In the absence of BSO (--No BSO--), a group of non-supplemented cells was pre-treated (12 h) in SFM and then divided into two sets; one set was incubated (12 h) in SFM without cadmium (No Cd/No Zn), while the other was exposed (12 h) to 500 nM CdCl_2_ in SFM (Cd). In absence of BSO as well (--No BSO--), a group of Zn-supplemented cells was pre-treated (12 h) in SFM with 10 µM ZnCl_2_ and then divided into two sets: one set was incubated (12 h) in SFM with 10 µM ZnCl_2_ (Zn), while the other was exposed (12 h) to 500 nM CdCl_2_ in SFM with 10 µM ZnCl_2_ (Zn + Cd). To test the effects of inhibition of glutathione synthesis, BSO was added to the SFM used for pre-treatment and CdCl_2_ exposure of non-supplemented and Zn-supplemented CPECs. In the presence of BSO (--BSO--), a group of non-supplemented cells was pre-treated (12 h) in SFM with 100 µM BSO and divided into two sets: one was incubated (12 h) in SFM with 100 µM BSO (BSO, No Cd/No Zn), while the other set was exposed (12 h) to 500 nM CdCl_2_ in SFM with 100 µM BSO (BSO + Cd). In the presence of BSO as well, (--BSO--), a group of Zn-supplemented cells was pre-treated (12 h) in SFM with 100 µM BSO and 10 µM ZnCl_2_ and then divided into two sets: one was incubated (12 h) with 100 µM BSO and 10 µM ZnCl_2_ (BSO + Zn); the other was exposed (12 h) to 500 nM CdCl_2_ in SFM 100 µM BSO and 10 µM ZnCl_2_ (BSO + Zn + Cd). All treatments were performed at 37 °C. The mRNA expression was analyzed in triplicate by quantitative real-time polymerase chain reaction (qRT-PCR); mRNA expression for each test gene was normalized to mRNA for rat β-actin (Actb) and GAPDH (Gapdh). Fold-induction was calculated as the ratio of normalized gene expression in Cd-exposed cells to that in time-matched controls. The mRNA was analyzed in triplicate in five separate culture preparations. Data are presented as means ± SE; *n* = 5; * *p* < 0.05 vs. Control; § *p* < 0.05 vs. Cd; # *p* < 0.05 Zn + BSO + Cd vs. BSO + Cd.

**Figure 7 ijms-22-08857-f007:**
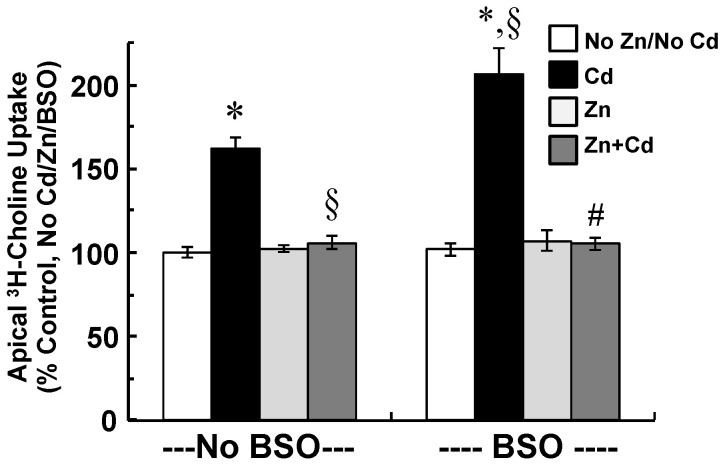
Effects of inhibition of glutathione (GSH) synthesis by *l*-buthionine sulfoximine (BSO) and Zinc supplementation on apical uptake of 10 µM [^3^H]choline in choroid plexus epithelial cells (CPECs) exposed for 12 h to 500 nM CdcCl_2_. CPECs were first incubated (48 h) in maintenance medium without Zinc; i.e., non-supplemented cells, or with 25 µM ZnCl_2_; i.e., Zn-supplemented cells. In the absence of BSO (--No BSO--), a group of non-supplemented cells was pre-treated (12 h) in serum-free medium (SFM) and then divided into two sets; one set was incubated (12 h) in SFM without cadmium (No Cd/No Zn), while the other was exposed (12 h) to 500 nM CdCl_2_ in SFM (Cd). In absence of BSO as well (--No BSO--), a group of Zn-supplemented cells was pre-treated (12 h) in SFM with 10 µM ZnCl_2_ and then divided into two sets: one set was incubated (12 h) in SFM with 10 µM ZnCl_2_ (Zn), while the other was exposed (12 h) to 500 nM CdCl_2_ in SFM with 10 µM ZnCl_2_ (Zn + Cd). To test the effects of GSH inhibition, BSO was added to the SFM used for pre-treatment and cadmium exposure of CPECs. In the presence of BSO (--BSO--): a group of non-supplemented cells was pre-treated (12 h) in SFM with 100 µM BSO and divided into two sets: one set was incubated (12 h) in SFM with 100 µM BSO (BSO, No Cd/No Zn), while the other was exposed (12 h) to 500 nM CdCl_2_ in SFM with 100 µM BSO (BSO + Cd). In the presence of BSO as well (--BSO--), a group of Zn-supplemented cells was pre-treated (12 h) in SFM with 100 µM BSO and 10 µM ZnCl_2_ and then divided into two sets: one was incubated (12 h) with 100 µM BSO and 10 µM ZnCl_2_ (BSO + Zn), while the other was exposed (12 h) to 500 nM CdCl_2_ in SFM with 100 µM BSO and 10 µM ZnCl_2_ (BSO + Zn + Cd). All treatments were performed at 37 °C. After experimental treatments, cells were rinsed and incubated (30 min, 37 °C) in artificial CSF (10 mM Tris-HEPES, pH 7.4) with 10 µM [^3^H]choline chloride ± 750 µM hemicholinium-3. Artificial CSF did not contain BSO, Zinc, or cadmium. Uptake was measured in triplicate in three different culture preparations (*n* = 3). Uptake was expressed as percentage of that in control; data are means ± SE; *n* = 3. * *p* < 0.05 vs. Control; § *p* < 0.05 vs. Cd; ^#^
*p* < 0.05 Zn + BSO + Cd vs. BSO + Cd.

**Table 1 ijms-22-08857-t001:** Elemental Zinc and elemental cadmium accumulation in choroid plexus epithelial cells (CPECs) exposed for 12 h to 500 nM CdCl_2_ and supplemented with Zinc.

Condition	Zinc, ng/mg Protein	Cadmium, ng/mg Protein
Control	333.82 ± 34.04	1.02 ± 0.47
Cd	348.99 ± 42.94	265.79 ± 11.79 *
Zn	556.61 ± 53.44 *	1.14 ± 0.05
Zn + Cd	463.26 ± 15.82 *	243.72 ± 8.62 *

CPECs were first incubated for 48 h in maintenance medium without Zinc; i.e., non-supplemented cells, or with 25 µM ZnCl_2_; i.e., Zn-supplemented cells. Non-supplemented cells were pre-treated (12 h) in serum-free medium (SFM) and then divided into two sets: one set was exposed for 12 h in SFM with 0 CdCl_2_ (Control), while the other was incubated for 12 h in SFM with 500 nM CdCl_2_ (Cd). In parallel, Zn-supplemented cells were pre-treated (12 h) in SFM with 10 µM ZnCl_2_ and then divided into two sets: one set was exposed for 12 h in SFM with 10 µM ZnCl_2_ alone (Zn), while the other was exposed for 12 h to 500 nM CdCl_2_ in SFM with 10 µM ZnCl_2_ (Zn + Cd). All incubations were at 37 °C. Cellular content of Zn and Cd was measured by inductively coupled plasma mass spectrophotometry (ICP-MS), normalized to cell protein, and expressed as ng/mg protein. Metal accumulation was measured in duplicate in three separate culture preparations. Data are means ± SE; *n* = 3 culture preparations. * *p* < 0.05 vs. Control.

**Table 2 ijms-22-08857-t002:** Effects of Zinc supplementation on glutathione (GSH) and glutathione sulfide (GSSG) concentrations and GSH:GSSG ratio in choroid plexus epithelial cells (CPECs) exposed to 500 nM CdCl_2_ for 12 h.

Condition	GSH, µM	GSSG, µM	GSH:GSSG Ratio
Control	5.20 ± 0.50	0.029 ± 0.010	318.4 ± 102.99
Cd	9.18 ± 0.27 *	0.207 ± 0.023 *	43.28 ± 5.89 *
Zn	5.25 ± 0.46	0.035 ± 0.010	273.73 ± 99.76
Zn + Cd	7.84 ± 0.87 *	0.121 ± 0.022 *	71.94 ± 14.97 *

CPECs were first incubated for 48 h in maintenance medium without Zinc; i.e., non-supplemented cells, or with 25 µM ZnCl_2_; i.e., Zn-supplemented cells. Non-supplemented cells were pre-treated (12 h) in serum-free medium (SFM) and then divided into two sets: one set was exposed for 12 h in SFM (Control), while the other was exposed for 12 h to 500 nM CdCl_2_ in SFM (Cd). In parallel, Zn-supplemented cells were pre-treated (12 h) in SFM with 10 µM ZnCl_2_ and then divided into two sets: one set was exposed for 12 h in SFM with 10 µM ZnCl_2_ (Zn), while the other was exposed for 12 h to 500 nM CdCl_2_ in SFM with 10 µM ZnCl_2_ (Zn + Cd). All incubations were at 37 °C. GSH and GSSG were analyzed by luminescence in triplicate in four different culture preparations (*n* = 4); GSH:GSSG ratios were calculated. Data are means ± SE; *n* = 4 culture preparations. * *p* < 0.05 vs. Control.

**Table 3 ijms-22-08857-t003:** Thirty-minute apical uptake of 10 µM [^3^H]choline in choroid plexus epithelial cells (CPECs) after 24 h pre-treatment with 5–100 µM ZnCl_2_ in serum-free medium (SFM).

Zinc Concentration, µM	[^3^H]Choline Uptake, pmol/mg Protein
0 (*Control*)	3274.27 ± 287.13
5	3373.58 ± 227.32
10	3288.23 ± 201.05
25	3343.04 ± 269.76
50	3339.53 ± 175.93
100	304.07 ± 96.00 *

CPECs were incubated (24 h) in SFM supplemented with 0 (Control) or 5–100 µM ZnCl_2_ at 37 °C. Cells were then rinsed and incubated (30 min, 37 °C) in Zinc-free artificial cerebrospinal fluid (10 mM Tris-HEPES, pH 7.4) with 10 µM [^3^H]choline chloride ± 750 µM hemicholinium-3. Thirty-minute [^3^H]choline uptake was expressed as pmol/mg protein. Uptake was measured in triplicate in three different culture preparations. Data are means ± SE; *n* = 3. * *p* < 0.05 vs. Control.

**Table 4 ijms-22-08857-t004:** Effects of Zinc supplementation and inhibition of glutathione (GSH) synthesis by *l*-buthionine sulfoximine (BSO) on GSH and glutathione sulfide (GSSG) concentrations and the GSH:GSSG ratio in choroid plexus epithelial cells (CEPCs) exposed for 12 h to 500 nM CdCl_2_.

Condition	GSH, µM	GSSG, µM	GSH:GSSG Ratio
Control	4.39 ± 0.727	0.050 ± 0.017	231.76 ± 121.66
Cd	9.35 ± 1.38 *	0.167 ± 0.012 *	46.90 ± 5.68 *
Zn	4.57 ± 0.70	0.045 ± 0.025	241.33 ± 130.55
Zn + Cd	8.11 ± 1.12 *	0.135 ± 0.030 *	54.09 ± 8.14 *
BSO	0.45 ± 0.50 *	0.073 ± 0.031	8.75 ± 3.25 *
BSO + Cd	0.23 ± 0.02 *^,^§	0.077 ± 0.023	9.37 ± 3.29 *
BSO + Zn	0.51 ± 0.07 *	0.074 ± 0.021	6.86 ± 2.26 *
BSO + Zn + Cd	0.33 ± 0.02 *^,^§	0.078 ± 0.025	8.09 ± 3.09 *

CPECs were first incubated (48 h) in maintenance medium without Zinc; i.e., non-supplemented cells, or with 25 µM ZnCl_2_; i.e., Zn-supplemented cells. In the absence of BSO, a group of non-supplemented cells was pre-treated (12 h) in serum-free medium (SFM) and then divided into two sets; one was incubated (12 h) in SFM without cadmium (Control), while the other was exposed (12 h) to 500 nM CdCl_2_ in SFM (Cd). In absence of BSO as well, a group of Zn-supplemented cells was pre-treated (12 h) in SFM with 10 µM ZnCl_2_ and then divided into two sets: one was incubated (12 h) with 10 µM ZnCl_2_ (Zn), while the other was exposed (12 h) to 500 nM CdCl_2_ in SFM with 10 µM ZnCl_2_ (Zn + Cd). In the presence of BSO, a group of non-supplemented CPECs was pre-treated (12 h) in SFM with 100 µM BSO and then divided into two sets; one was incubated (12 h) in SFM with 100 µM BSO (BSO), while the other was exposed (12 h) to 500 nM CdCl_2_ in SFM with 100 µM BSO (BSO + Cd). In the presence of BSO as well, a group of Zn-supplemented cells was pre-treated (12 h) in SFM with 100 µM BSO and 10 µM ZnCl_2_ and then divided into two sets: one set was incubated (12 h) with 100 µM BSO and 10 µM ZnCl_2_ (BSO + Zn); the other was exposed (12 h) to 500 nM CdCl_2_ in SFM 100 µM BSO and 10 µM ZnCl_2_ (BSO + Zn + Cd). All treatments were performed at 37 °C. GSH and GSSG concentrations were analyzed by luminescence in triplicate in four different culture preparations; ratios of GSH to GSSG (GSH:GSSG ratio) were subsequently calculated. Data are means ± SE; *n* = 4. * *p* < 0.05 vs. Control; § *p* < 0.05 vs. BSO alone.

**Table 5 ijms-22-08857-t005:** Forward and reverse primer sequences used to analyze gene expression in rat choroid plexus epithelial cells and isolated rat choroid plexus tissues, listed by respective gene name.

Gene	GeneBank ID	Forward Primer (5–3′)	Reverse Primer (3′–5′)
Actb	NM_031144	ATGGTGGGTATGGGTCAG	TACTTCAGGGTCAGGATGC
Gapdh	NM_017008	ATGACTCTACCCACGGC	ACTCAGCACCAGCATCA
Gclc	NM_012815	GCTTTCTCCTACCTGTTTCTTG	TGGCAGAGTTCAGTTCCG
Gclm	NM_017305	TGTGATGCCACCAGATTTGA	TGGAAACTTGCCTCAGAGAG
Hmox1	NM_012580	ACCCCACAAGTTCAAACAG	CCTCTGGCGAAGAACTCTG
Hspa4	NM_153629	ATGGGGGACAAGTCGGA	GTGGGGATGGTGGAGTT
Mt1	NM_138826	CACCGTTGCTCCAGATTCA	CAGCAGCACTGTTCGTCA

**Table 6 ijms-22-08857-t006:** Lactate dehydrogenase (LDH) release in choroid plexus epithelial cells exposed for 12 h to 500 nM CdCl_2_ with Zinc supplementation and inhibition of glutathione synthesis by *l*-buthionine sulfoximine (BSO).

Condition	LDH Release (% Maximal Release)
Control	11.39 ± 0.76
Cd	14.09 ± 1.54
Zn	10.34 ± 0.20
Zn + Cd	10.49 ± 0.22
BSO	10.89 ± 0.34
BSO + Cd	12.85 ± 1.01
BSO + Zn	10.55 ± 0.85
BSO + Zn + Cd	10.61 ± 0.29

Cells were first incubated (48 h) in maintenance medium without Zinc; i.e., non-supplemented cells, or with 25 µM ZnCl_2_; i.e., Zn-supplemented cells. In the absence of BSO, a group of non-supplemented cells was then pre-treated (12 h) in serum-free medium (SFM); thereafter, cells were divided into two sets: one set was incubated (12 h) in SFM with 0 CdCl_2_ (Control), and the other was exposed (12 h) to 500 nM CdCl_2_ in SFM (Cd). In the absence of BSO as well, a group of Zn-supplemented cells was pre-treated (12 h) in SFM with 10 µM ZnCl_2_ and then divided into two sets: one was incubated (12 h) with 10 µM ZnCl_2_ (*Zn*), while the other was exposed (12 h) to 500 nM CdCl_2_ in SFM with 10 µM ZnCl_2_ (Zn + Cd). In the presence of BSO, a group of non-supplemented CPECs was pre-treated (12 h) in SFM with 100 µM BSO and then divided into two sets: one was incubated (12 h) in SFM with 100 µM BSO (BSO), while the other was exposed (12 h) to 500 nM CdCl_2_ in SFM with 100 µM BSO (BSO + Cd). In the presence of BSO as well, a group of Zn-supplemented cells was pre-treated (12 h) in SFM with 100 µM BSO and 10 µM ZnCl_2_ and then divided into two sets: one was incubated (12 h) with 100 µM BSO and 10 µM ZnCl_2_ (BSO + Zn), while the other was exposed (12 h) to 500 nM CdCl_2_ in SFM with 100 µM BSO and 10 µM ZnCl_2_ (*BSO + Zn + Cd*). All treatments were performed at 37 °C. LDH was measured in triplicate in three individual culture preparations. Data are means ± SE (*n* = 3). There were no differences in LDH release among the experimental and control treatments (*p* > 0.2).

## Data Availability

The data presented in this study are available upon request from the corresponding author.

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
