# Peer review of "GSH and Zinc Supplementation Attenuate Cadmium-Induced Cellular Stress and Stimulation of Choline Uptake in Cultured Neonatal Rat Choroid Plexus Epithelia"

_ijms, 2021, doi:10.3390/ijms22168857_

Round 1

Reviewer 1 Report

The article “GSH and zinc supplementation attenuate cadmium-induced cellular stress and stimulation of choline uptake in cultured neonatal rat choroid plexus epithelia” refers to the capacity of CP cells to sequester metals that may be harmful to brain function. This feature of the CP has not been much investigated, what enhances the interest of the article.

Overall, the article is well written but some sentences are confusing and too long enhancing the complexity of the text that is often hard to follow.

A complementary analysis that should be performed is that of the permeability of the CP epithelia upon CdCl2 and ZnCl2 exposure through the analysis of tight junctions’ proteins like E-cadherin. This would further sustain the effects of these metals on the integrity of the CP epithelia.

A direct method to measure ROS in these experiments would be of interest to further complement and support the conclusions on the protective effects of Zn against ROS.

Introduction

The introduction contextualizes the relevance of the study although some sentences require revision to improve its reading: the sentence between lines 52 and 56 is very long and confusing; the same applies to the sentence between lines 72-75.

In 97-100 there is no reference to the ex vivo experiments

Results:

Since the material and methods section is placed after the discussion, the results section requires a better description of the experiments that yielded the results, and an explanation of the rational behind the experiments and of the parameters analysed leaving  the details for the methods section.

Of notice, some abbreviations, only defined in methods  are already used in this section (eg.  line 107- ICP-MS)

In section 2.1 the description of the experiments is confusing and it is difficult to correlate with the results and with the experiments described in the material and methods section.

Table 1-it is unclear whether the experiments were repeated 3 times or if it was made once using triplicates. This clarification should be checked throughout the text.

In section 2.3, there is no description of  the time exposure experiments referred in this section, which led to the results shown in figure 1 The methods section does not mention these experiments either.

The legends of figure 3 should mention the exposure time to Cd

It is  unclear why 24h of Cd exposure were chosen for the experiments in Fig 3 instead of one of the exposure times tested in fig 1

Figure 5-the much higher fold increase in Mt1 compared to the other expression of the other genes impairs the perception of their fold changes. An alternative way of presenting these data should be used to highlight the differences between the expression of the other genes

Discussion

It is unclear why attenuation of choline uptake is a measure of cytoprotection against cadmium (lines 298-299)

LDH results are not discussed, since there are no differences in LDH how is that zinc conveys cytoprotection?

Ex vivo in vivo data are not compare or discussed

Methods

Ex vivo vs vitro- These designations are not consistently used throughout the text.

Why ex vivo experiments were only used to compare gene expression, and not the rest of the parameters analysed?

Experiments with serial concentrations of zinc are not presented in results or supplementary data (line 491)

Description of experiments with ZnCl2 pre-treatments, and combinations of ZnCl2 and CdCl2 are described in a confusing way and it is difficult to correlate with the experiments presented in the results section.

The time exposure to CdCl2 experiments are not described in the methods section

Author Response

Responses to Reviewer No. 1:

General Remarks to Authors:

The article “GSH and zinc supplementation attenuate cadmium-induced cellular stress and stimulation of choline uptake in cultured neonatal rat choroid plexus epithelia” refers to the capacity of CP cells to sequester metals that may be harmful to brain function. This feature of the CP has not been much investigated, what enhances the interest of the article.

**Response:  We thank the reviewer for their time in reviewing the manuscript and for their insightful and valuable suggestions, which have helped to greatly improve the manuscript.

Overall, the article is well written but some sentences are confusing and too long enhancing the complexity of the text that is often hard to follow.

**Response:  We have rewritten numerous sections of the text, particularly in the Introduction.  We trust these edits will have made the manuscript easier to follow. 

A complementary analysis that should be performed is that of the permeability of the CP epithelia upon CdCl2 and ZnCl2 exposure through the analysis of tight junctions’ proteins like E-cadherin. This would further sustain the effects of these metals on the integrity of the CP epithelia.

*Response:  A complementary analysis of modulation of the permeability (e.g., transepithelial electrical resistance, paracellular flux of solute or extracellular marked) of the CP epithelia and variations in tight junction proteins, such as E-cadherin, claudin and occludin, in response to CdCl2 and ZnCl2 exposure would make for an extremely insightful study.  However, this is beyond the scope of the present study.

A direct method to measure ROS in these experiments would be of interest to further complement and support the conclusions on the protective effects of Zn against ROS.

**Response:  We agree that direct measurement of ROS (reactive oxygen species) would be of interest and further complement our conclusions that zinc functions to protect against ROS and cellular stress.  Although that is beyond the scope of the present study, but could be pursued in future investigations that focus more exclusively on the pro-antioxidant role of zinc.  

Specific Remarks regarding the Introduction

The introduction contextualizes the relevance of the study although some sentences require revision to improve its reading: the sentence between lines 52 and 56 is very long and confusing; the same applies to the sentence between lines 72-75.

**Response: We have revised the wording of the sentence between lines 52 and 56 and converted the one long sentence to two sentences. That portion of text now reads:

Investigation of the potential effects of cadmium on function, such as solute transport, is limited. We had shown in primary cultured neonatal rat CP epithelial cells (CPECs) that subchronic exposure to 500 nM cadmium induced oxidative cellular stress response and stimulated apical uptake of choline [17]. Choline is a model substrate for organic cation transporter-2 (OCT-2, SLC22A2) localized to the apical membrane [18] and precursor to the neurotransmitter acetylcholine (ACh).

Other long sentences were also converted to two or three shorter sentences to improve the clarity of the text and better communicate our intended meaning.

The sentence between lines 61-64 was also converted to two sentences:  “Stimulation of choline uptake at the apical membrane of CPECs is analogous to increased removal of choline from the CSF compartment in situ. This which could potentially limit central availability of choline and ACh. “

The Reviewer also specifically indicated the sentence between lines 72-75 of the original manuscript was long and confusing.  We have edited that entire paragraph that begins “Glutathione (GSH) is the most abundant intracellular thiol …” and starts lines 66/67 in the revised manuscript.  It now reads:   “Glutathione (GSH) is the most abundant intracellular thiol and predominant intracellular antioxidant [22]. GSH is readily available and can directly bind cadmium GSH.  As such, it is considered the first line of defense against cadmium-induced oxidative cellular stress [23,24]. However, cadmium can deplete intracellular GSH [13,25]. We showed in CPECs that a precursor to GSH, n-acetylcysteine, attenuated induction of Hsp70 and stimulation of apical choline uptake by cadmium. Conversely, l-buthionine-sulfoximine (BSO), an inhibitor of the rate-limiting enzyme in GSH synthesis glutamate cysteine ligase (GCL), enhanced both Hsp70 induction and stimulation of choline uptake [17]. Thus, a cellular stress response was mounted in response to an oxidative stress induced by cadmium. This suggested GSH might be critical to the epithelium’s adaptation to cadmium-induced cellular stress. GCL is highly expressed in CP, and GSH synthesis is integral to both gamma-glutamyl cycling and amino acid transport [26] and phase II metabolism and drug clearance from CSF [27]. In CP the antioxidant GSH system may serve to regulate accumulation of reactive oxygen species (ROS) produced by basal metabolism and ROS generated upon induction of oxidative stress, such as by elevated levels of H2O2 in CSF [28]. However, the role of GSH in adaptation to cellular stress elicited by cadmium or other contaminant metals sequestered by CP has not been fully elucidated. “

To better communicate that one mechanism by which zinc functions to support the antioxidant defenses, the sentence that began “Furthermore, the redox inert metal .. “ in the original manuscript, we modified the wording.  That sentence between lines 85-86 in the revised manuscript t now begins: “Furthermore, zinc is a redox inert metal that promotes cellular antioxidant mechanisms, … “ 

In the last sentence of the next to last paragraph of the Introduction, we omitted the phrase “of 17%, as that information was not essential.   

In 97-100 there is no reference to the ex vivo experiments

**Response:  In the revised manuscript, the final statement of the Introduction (lines 97-100) now includes a reference to the experiments in which isolated lateral and fourth choroid plexus were exposed to cadmium. “Using primary cultured neonatal rat CPECs and isolated CP tissues exposed to submicromolar cadmium, our aim was to characterize the potentially protective roles of GSH and zinc supplementation in the adaptive stress response to cadmium.”

Specific Remarks regarding the Results

Since the material and methods section is placed after the discussion, the results section requires a better description of the experiments that yielded the results, and an explanation of the rational(e) behind the experiments and of the parameters analysed leaving the details for the methods section.

**Response:  We agree completely with the reviewer.  If the Materials/Methods section follows the Discussion (which is the conventional order for The Journal), the Results section should include better descriptions of the experiment treatments and the rationale for the experiments.  In the revised manuscript, we have explained the rationale or purpose of each experiment and the parameters that were analyzed – beginning with the experiments in section 2.1.  In addition, as suggested by Reviewer #2, we have included a schematic diagram in Figure 7 that outlines the experimental treatments in the Materials/Methods-section 4.4.  When describing experimental treatments in the Results, we now refer to the specific treatment in Figure 7.  In the first paragraph of the Results, we also state:  “Experimental treatments of CPECs and isolated CP tissues implemented in this study are described in detail in the Material and Methods (section 4.4) and outlined in the schematic diagram in Figure 7. However, reporting of experimental results will include succinct descriptions of the respective treatments and references to Figure 7. “   In addition, the revised table and figure legends also now include more detailed descriptions of experimental treatments (as suggested by Reviewer #2).  The inclusion of these more detailed descriptions of experimental treatments and rationale has made the manuscript noticeably longer.  However, this should promote clearer understanding of both the experiment design and the interpretation of our findings.

Of notice, some abbreviations, only defined in methods are already used in this section (eg. line 107- ICP-MS).

**Response:  We thank the reviewer for bringing this to our attention and apologize for this oversight.  We have defined the abbreviation, ICPMS (“inductively coupled plasma mass spectrometry (ICP-MS)” in the Results (line 133-135) and other abbreviations that were not defined, e.g., SFM: serum-free medium.  As suggested by Reviewer #2, we also have defined abbreviations in the revised table and figure legends.

In section 2.1 the description of the experiments is confusing and it is difficult to correlate with the results and with the experiments described in the material and methods section.

**Response:  We have now included a clearer description of the experimental treatments and rationale for measurement of Cd and Zn accumulation in choroid plexus epithelial cells (CPECs) in section 2.1-line 114-131.  As explained in our responses above, in the first paragraph of the Results, we now state:  “Experimental treatments of CPECs and isolated CP tissues implemented in this study are described in detail in the Material and Methods (section 4.4) and outlined in the schematic diagram in Figure 7. However, reporting of experimental results will include succinct descriptions of the respective treatments and references to Figure 7. “  

Table 1-it is unclear whether the experiments were repeated 3 times or if it was made once using triplicates. This clarification should be checked throughout the text.

**Response:  Accumulation of Cd and Zn was measured by ICP-MS in duplicate (duplicate measures) in three separate culture preparations, i.e., n = 3.  We have clarified this in the legend for Table 1.  We have included the number of replicate measures and the number of cultures preparations in which a given parameter was measured in the table and figure legends.  We also include this information in the Material/Methods when describing a given method of analysis or assay. 

In section 2.3, there is no description of the time exposure experiments referred in this section, which led to the results shown in figure 1 The methods section does not mention these experiments either.

**Response:  We thank the reviewer for bringing this to our attention and apologize for this oversight in omitting this information in the original manuscript.  We have included a more detailed description of treatment of CPECs to examine mRNA and protein expression for stress proteins and GLC subunits during the 12-h time course exposure to 500 nM CdCl2 in the text in section 2.3 of the Results, the legend for Figures 1 and 2, section 4.4 of the Material/Methods, and the schematic diagram Figure 7.

The legends of figure 3 should mention the exposure time to Cd.

**Response:  We apologize for not including this information in the original manuscript.  Isolated choroid plexus tissues were exposed to 500 nM CdCl2 in serum-free medium for 24 hours at 37ËšC.  We have included a detailed description of treatment of isolated choroid plexus tissues in the text of the Results-section 2.3 in, Figure 3 legend, and Material/Methods-section 4.4, and Figure 7.  

It is unclear why 24h of Cd exposure were chosen for the experiments in Fig 3 instead of one of the exposure times tested in fig 1

**Response:  We now briefly explain this in the text of the Results-section 2.3 (lines 256-258).  Given the complex organization of epithelial cells and vascular tissues (endothelial, smooth muscle) of the intact choroid plexus, we extended the exposure time to 24 h versus the 12-h exposure time for single-layered CPECs.  In our prior study [Ref. #17], CPECs exposed to 500 nM CdCl2 for up to 24 h remained responsive to cadmium treatment, i.e., apical uptake of [3H]choline was still markedly stimulated. 

Figure 5-the much higher fold increase in Mt1 compared to the other expression of the other genes impairs the perception of their fold changes. An alternative way of presenting these data should be used to highlight the differences between the expression of the other genes

**Response:  The reviewer has made a valid suggestion.  However, we wish to continue to graphically present these data in the original graph.  We have directly reported the quantitative comparisons of the fold-induction Mt1 mRNA induction versus induction of mRNA expression for the other stress proteins and GCL subunits. 

Specific Remarks regarding the Discussion

It is unclear why attenuation of choline uptake is a measure of cytoprotection against cadmium (lines 298-299) –

*Response:  This comment and our response are related to the Reviewer’s next comment and our respective response.  Based on the lack of differences in LDH release among the control and experimental treatments, treatment with Cadmium alone or in combination with BSO to inhibit GSH synthesis did not elicit marked cytotoxicity.  However, even in the absence of marked cytotoxicity, such as with cadmium exposure alone, CPECs mounted a cellular stress response – as indicated by marked induction of Hsp70/HSpa4, HO/Hmox1 and metallothionein mRNA (Mt1.  Moreover, in response to cadmium exposure with BSO treatment to inhibit GSH synthesis, induction of these stress proteins was even greater.  Thus, while exposure to 500 nM CdCl2 for 12 h might not elicit cytotoxicity, cell biology has been disrupted and there is some degree of protein unfolding.  In the context of the cellular stress, apical choline transport markedly increases; moreover, as cellular stress is exacerbated, stimulation of transport is further enhanced.  An increase in function response to a given stimulus is not always beneficial.  Solute transport, either into or out of cerebrospinal fluid, by the choroid plexus epithelium is critical to central neural homeostasis.  Our early work in CPECs [Ref #74] showed that apical choline transport was a electrosensitive secondary active process functional coupled to Na,K-ATPase. Thus, apical choline transport is coupled to cellular metabolism.  In our view of the biology of choroid plexus, stress-induced increases in active solute transport is a liability.  Thus, given that zinc may minimize cellular stress and preserve baseline transport function when choroid plexus is exposed to and has accumulated a toxic metal, we consider zinc to be cytoprotective.  Moreover, our data indicate that zinc supplementation did not impair the capacity of CPECs to accumulate cadmium. Thus, zinc is not only protective of choroid plexus biology, but could potentially protect this epithelium as it still sequesters a toxic metals and can then continue to protect the brain from their neurotoxic effects. 

LDH results are not discussed, since there are no differences in LDH how is that zinc conveys cytoprotection?

*Response:  As suggested by Reviewer #1, the data for LDH release in response to cadmium exposure, zinc supplementation, inhibition of GSH synthesis with BSO, and combination of these treatments was moved to the Materials/Methods section. The lack of differences in LDH release among the control and experimental treatments indicated the treatments did not elicit marked cytotoxicity.  However, even in the absence of marked cytotoxicity, such as with cadmium exposure alone, CPECs mounted a cellular stress response – as indicated by marked induction of Hsp70/HSpa4, HO/Hmox1 and metallothionein mRNA (Mt1) in the present study and in our prior study.  Moreover, in response to cadmium exposure with BSO treatment to inhibit GSH synthesis, induction of these stress proteins was even greater.  As cited in the Discussion, the induction of Hsp70 is triggered by protein unfolding and proportional to severity of cellular stress [Ref #56,57].  In context of this cellular stress induced by cadmium in CPECs, choline uptake was stimulated and stimulated to an even greater extent when GSH synthesis is inhibited by BSO, which exacerbated cellular stress.  Thus, despite there is no marked increase in LDH release or marked cytotoxicity, the epithelial cell has been subjected to cellular insult and stress.  Increased gene expression as well as increase protein synthesis/expression is biologically costly.  However, when cells are supplemented with zinc prior to cadmium exposure, the extent of cellular stress is attenuated, i.e., stress protein induction is attenuated and stimulation choline transport is abated.  This effect was also observed when cellular stress was exacerbated with inhibition of GSH synthesis with BSO.  Thus, we consider zinc to be cytoprotective.  I have added this statement: “Our data indicate that zinc did not impair the capacity of CPECs to accumulate cadmium. Thus, zinc is not only protective of choroid plexus biology, but could potentially protect this epithelium as it still sequesters a toxic metals and continues to protect the brain from their neurotoxic effects.” To the conclusion of the manuscript.

Ex vivo in vivo data are not compare or discussed

*Response:  We have included discussion of the qualitatively similar induction of mRNA for Hspa4, Hmox1, Mt1, Gclm, and Gclc in isolated CP tissue exposed in vitro as observed in CPECs.

Specific Remarks regarding the Methods

Ex vivo vs vitro- These designations are not consistently used throughout the text.

Why ex vivo experiments were only used to compare gene expression, and not the rest of the parameters analysed?

**Response:  We thank the reviewer for bring this inconsistency to our attention.  We no longer use the term “ex vivo” in the revised manuscript.  We refer to the exposure to isolated choroid plexus tissues only as an “in vitro” exposure” in the text of the Results, Figure 3 legend, Material/Methods. 

Experiments with serial concentrations of zinc are not presented in results or supplementary data (line 491)

*Response:  This involved a series of preliminary trials of treated CPECs with 5-100 µM ZnCl2 in for 48 h to 72 h.  We believe the data presented already in the manuscript sufficiently substantiates the validity of the selection of the concentrations of ZnCl2 used in our current study.  These data are those for the effects of 24 h zinc pretreatment in serum-free medium on choline uptake, as well as the collection data that indicate the protocol for zinc supplementation is not toxic in CPECs, does not impair the ability to accumulation cadmium, does not impair GSH chemistry, does not (be itself) modulate choline uptake, and is cytoprotective.

Description of experiments with ZnCl2 pre-treatments, and combinations of ZnCl2 and CdCl2 are described in a confusing way and it is difficult to correlate with the experiments presented in the results section.

**Response:  In the Materials/Methods-section 4.4 of the revised manuscript we have included more detailed description of experimental protocols for individual and combined treatments of ZnCl2 supplementation, CdCl2 exposure, and l-buthionine sulfoximine (BSO).  As suggested by Reviewer #2, we have included a new figure - Figure 7, which presents a simple schematic diagram outlining the various experimental treatments 

The time exposure to CdCl2 experiments are not described in the methods section

**Response: We apologize for the oversight.  In the revised manuscript we have described the protocol for the 12-h time-course exposure to 500 nM CdCl2, in the text of Results, the legend for Figures 1 and 2, as well as in the Materials/Methods-section 4,4 and Figure 7. “ CPECs were first incubated for 48 h in complete maintenance medium without zinc supplement, i.e., non-supplemented cells. These cells were subsequently pretreated for 12 h in SFM and then exposed to 0 (Control) or 500 nM CdCl2 (Cd) in SFM for up to 12 h (Fig 7B). At 3, 6, 9, and 12 h representative cadmium-exposed and time-matched control cells were collected for analysis of mRNA and protein expression by quantitative real-time polymerase chain reaction or immunoblot analysis, respectively. “

Reviewer 2 Report

Comments and Suggestions for Authors

Francis Stuart and Villalobos present in this manuscript impact of Zinc supplementation on cadmium-induced cellular stress and elevation of choline uptake in choroid plexus epithelial cells. The authors provide quite complex in vitro studies for which results and material and methods section need to be improved.

Importantly, the authors present results previously described in the master thesis of SD Francis Stuart, titled “The pro-antioxidant role of zinc supplementation in cadmium-treated choroid plexus”, which is available online at Texas A&M University Libraries. I understand the need to publish the results in the form of a scientific paper, but the information that the presented results constitute a master thesis should be included on the title page.

Major/Minor concerns:

Abstract: Line 23 looks like the word "uptake" is missing after choline;

  1. Results:

Note regarding all tables and figures: descriptions should be supplemented with explanations of abbreviations so that each table and figure taken from the context of the publication is as informative as possible;

Line 126: looks like the word "respectively" is missing after … 1.75- and 7-fold (p < 0.05)...;

In Figures 1 and 2, the colors of the bars should be the same when referring to the same factors tested at both the mRNA and protein levels.

Figure 3 has an erroneous description of the y-axis which should sound like a “Relative mRNA expression” since the authors did not specify the copy number of the genes.

I would suggest moving Table 5 to the Materials and Methods section as this is only a preliminary analysis that confirms the absence of cytotoxic effects of all treatments combinations and there is no discussion of this result.

  1. Materials and Methods

Is the protocol described in lines 474-477 to remove fibroblasts from the culture? If so, this information should be added to the text. In addition, I have a question whether CPEC cells from newborn rats do not require coating as it happens in the case of CPEC obtained from several weeks old animals?

Line 488: please provide references for CdCl2 dose selection;

Line 500: please provide references for BSO dose selection or an explanation on what basis it was selected;

Section 4.4.: I would like to encourage the authors to draw a simple diagram of the experiments that would facilitate the perception of this work. This should include all experiments on both CPEC and intact CP and include the number of repetitions.

Section 4.5.: please provide references for ICP-MS as a method for determination of zinc and cadmium accumulation;

Section 4.7., Line 559: please provide the name of the manufacturer of the nanospectroscopic measurement device;

Section 4.7., Line 564: please provide the information on how the reference genes were selected;

Section 4.8.: please provide the catalog numbers of used antibodies;

Section 4.11.: please provide the name and manufacturer of statistical software and the information on what kind of posthoc tests were used for each analysis;

General remark: I would like to suggest changing the abbreviations of the studied genes to standardize nomenclature, I would suggest using the HUGO Gene Nomenclature Committee, available at https://www.genenames.org

Author Response

Responses to Reviewer No. 2:

Introductory Remarks to Authors:

Francis Stuart and Villalobos present in this manuscript impact of Zinc supplementation on cadmium-induced cellular stress and elevation of choline uptake in choroid plexus epithelial cells. The authors provide quite complex in vitro studies for which results and material and methods section need to be improved.

**Response: We thank the reviewer for their time and their suggestions for improving the manuscript, particular in terms of including more detailed descriptions of the complex experimental treatments implemented in our study.  We have made this and other changes in the manuscript that have markedly improved the manuscript.

Importantly, the authors present results previously described in the master thesis of SD Francis Stuart, titled “The pro-antioxidant role of zinc supplementation in cadmium-treated choroid plexus”, which is available online at Texas A&M University Libraries. I understand the need to publish the results in the form of a scientific paper, but the information that the presented results constitute a master thesis should be included on the title page.

**Response: We have consulted The Journal editorial staff regarding The Journal’s guidelines for including this information.  We were told that this information is not to be included on the Title page.  Thus, in the revised manuscript the statement “ This work was conducted in partial fulfillment for the degree of master of science at Texas A&M University in the subject of toxicology by Samantha D. Francis Stuart. “ remains in the Acknowledgements section.  

Specific Remarks regrading the Abstract

Line 23 looks like the word "uptake" is missing after choline;

**Response: The word “uptake” was indeed missing in the phrase “choline uptake” and has been inserted after the “choline” in the revised manuscript.  Making this correction the Abstract increased the word count to 201.  The word limit for abstracts in IJMS is 200.  Therefore, we have deleted the prepositional phrase “with BSO” at the end of the very same sentence.  The new word count is 199.  This deletion does not change the meaning of the sentence or our intended interpretation. 

Specific Remarks regrading Results

Note regarding all tables and figures: descriptions should be supplemented with explanations of abbreviations so that each table and figure taken from the context of the publication is as informative as possible;

**Response:  In the revised manuscript, we have defined abbreviations in the legends of all tables and figures.  Additionally, as per suggestion by both reviewers, we have included more detailed description of experiments in each legend. These changes should make the legends as informative as possible, such that tables and figures can ‘stand alone’ ; that is the reader will not need to refer to the text.  Although these changes have increased the length of the manuscript, this fundamental albeit global change has improved the clarity of our experimental approach and findings.

Line 126: looks like the word "respectively" is missing after … 1.75- and 7-fold (p < 0.05)...;

**Response:  In the revised, manuscript (lines 172-173), “respectively” has been inserted after the phrase “1.75- and 7-fold (p < 0.05) ...”. 

In Figures 1 and 2, the colors of the bars should be the same when referring to the same factors tested at both the mRNA and protein levels.

**Response:  We thank the reviewer for this suggestion.  The color or pattern for the bar that represents the expression of a given gene in Figure 1 is now the same color or pattern as the bar that represents the expression of the corresponding protein in Figure 2.  

Figure 3 has an erroneous description of the y-axis which should sound like a “Relative mRNA expression” since the authors did not specify the copy number of the genes.

**Response:  We apologize for this error.  The y-axis title for the graph in Figure 3 has been changed to “Relative mRNA Expression”. 

I would suggest moving Table 5 to the Materials and Methods section as this is only a preliminary analysis that confirms the absence of cytotoxic effects of all treatments combinations and there is no discussion of this result.

**Response:  We thank the reviewer for this suggestion.  In the revised manuscript, Table 5 that presents date for lactate dehydrogenase (LDH) release from CPECs treated with combinations of cadmium, zinc and BSO has been moved to the Materials/Methods-section 4.10.  The LDH assay was used to screen for potential cytotoxicity of cadmium, zinc and BSO treatments.

Specific Remarks regrading Materials and Methods

Is the protocol described in lines 474-477 to remove fibroblasts from the culture? If so, this information should be added to the text. In addition, I have a question whether CPEC cells from newborn rats do not require coating as it happens in the case of CPEC obtained from several weeks old animals?

**Response:  We appreciate this suggestion.  In the revised manuscript, the description of the protocol for the isolation and primary culture of choroid plexus epithelial cells (CPECs) in the Materials/Methods-section 4.3 now includes brief explanations of the role of the preplating step and the purpose of and basis for using minimum essential medium with D-valine to minimize fibroblast contamination in the final yield of CPECs. Lines 669-672: “Cells were suspended in penicillin-supplemented DMEM/F12 with 10% Nu-Serum IV and pre-plated in a 35-mm Petri dish for 3.5 h at 37ËšC (humidified 95% air/5% CO2); during this time extraneous fibroblasts to attach, thereby minimizing fibroblast contamination of the CP epithelial cell culture. “  Also, lines 675-678: “The base of plating medium is MEM with d-valine substituted for l-valine; d-valine is poorly metabolized by fibroblasts, which limits fibroblast survival and proliferation; thus, fibroblast contamination of CP epithelial primary cultures [75].”  Reference #75, Lazzaro et al. 1992, is now cited to support the use of MEM with d-valine. 

**In response to Reviewer #2’s question: “… I have a question whether CPEC cells from newborn rats do not require coating as it happens in the case of CPEC obtained from several weeks old animals?”, I, Dr. Villalobos, offer this answer:  My experience in primary culture of choroid plexus epithelial cells (CPECs) is limited to culturing cells from neonatal rats. [I have not attempted to culture CPECs from a rat older than 5 days.]  We have not needed to coat the sterile polystyrene tissue culture plates that we have used in the final plating of CPECs in our protocol for primary culture of epithelial cells from neonatal rat choroid plexus.  We wait 72 hours after the (final) plating of CPECs to remove the plating medium and any unattached cells and then add maintenance medium.  As a rule, we view the cells by inverted light microscopy before changing out the initial plating medium.  A high-fraction epithelial cells have attached to the plating surface by 72 hours, and by 6 days, the cells have formed a confluent monolayer.  However, when plating cells on glass, e.g., chamber slides, such as for experiments involving viewing by epi- or confocal fluorescence microscopy, we will coat the slides with a thin layer of poly-D-lysine to increase the initial adhesion and attachment of CPECs to the glass surface. 

Line 488: please provide references for CdCl2 dose selection;

**Response: We have explained the basis of our selection of 500 nM CdCl2 in lines 686-691 in section 4.4 of Materials/Methods:  “To study the effects of cadmium in CPECs, cells were exposed to 500 nM CdCl2 for 12 h. We selected this concentration and duration of cadmium exposure, based our prior investigation of time-dependent and concentration-dependent exposure of CPECs to 0-1000 nM CdCl2 [17]. Twelve-hour exposure to 500 nM CdCl2induced a cellular stress response and marked stimulation in apical choline uptake, but did not increase lactate dehydrogenase release, i.e., caused minimal cytotoxicity.”

Line 500: please provide references for BSO dose selection or an explanation on what basis it was selected;

**Response:  We had used 100 µM BSO in our prior study on CPECs [Ref #17].  Studies using various cell types to investigate the role of GSH in cell biology or toxicology, BSO has been used in concentration as high 10 mM (Kang and Enger, 1988).  However, we based our selection of 100 µM BSO on work by Suthanthiran et al. (PNAS - 1990), which indicate BSO was effective 10 µM, 100 µM and 1 mM.  We opted for 100 µM BSO.  Data in Table 4 indicate that concentration is sufficient to decrease GSH levels in the presence or absence of cadmium exposure. We have cited this paper in the text of section 4.4 of the Material/Methods.

Section 4.4.: I would like to encourage the authors to draw a simple diagram of the experiments that would facilitate the perception of this work. This should include all experiments on both CPEC and intact CP and include the number of repetitions.

**Response:  We thank the author for the suggestion.  We have constructed such a (new) figure - Figure 7, which is presented in the Materials/Methods-section 4.4.  Figure 7 is a schematic outline of the experimental treatments of CPECs and isolated choroid plexus tissue.  Neither the figure nor its legend includes the number of repetitions; the figure is simple, but rather full.  Instead, we have included the number of measures (e.g., duplicate, triplicate) and the number of separate culture preparations or sets of tissues on which a given experiment was performed is reported in the respective table or figure legend.   Also, as suggested by Reviewer #1, we have included more detailed descriptions of experimental treatments in the Materials/Methods-section 4.4, but also in the Results.  Reviewer #1 commented, “Since the material and methods section is placed after the discussion, the results section requires a better description of the experiments that yielded the results, and an explanation of the rational(e) behind the experiments and of the parameters analysed leaving the details for the methods section.”   The Materials/Methods section follows the Discussion, which is the conventional order for The Journal.  Thus, we have included explanations for the rationale or purpose for each experiment and the parameters that were analyzed – beginning with the experiments in section 2.1 of the Results.  Although including these improved descriptions of experimental treatments and rationale, the revised manuscript is significantly longer.  However, these changes offer the reader a clearer understanding of the experiments, as well as our findings and interpretations.

Section 4.5.: please provide references for ICP-MS as a method for determination of zinc and cadmium accumulation;

**Response:  We now include the following explanation in the Material/Methods-section 4.5:  “The protocol for elemental metal analysis in cultured cells by ICP-MS is the standard analysis protocol used in Trace Element Research Laboratory (College of Veterinary Medicine & Biomedical Sciences, Texas A&M University-College Station), which performed the analysis. This same protocol was used to determine cellular cadmium content in CPECs in our prior investigation of cadmium-induced stress responses in CPECs [17].”

Section 4.7., Line 559: please provide the name of the manufacturer of the nanospectroscopic measurement device;

**Response:  In the Material/Methods-section 4.7, we have included the name of the manufacturer of the nanospectroscopic measurement device, as well as city and state of the manufacturer: NanoDrop ® ND-100, Thermo Scientific, Waltham, MA  

Section 4.7., Line 564: please provide the information on how the reference genes were selected;

**Response: Reference genes Actb and Gapdh were used as reference genes to normal expression of the genes of interest in this study. Actb and Gapdh are considered ‘housekeeping genes’.  We use these the present study, because mRNA of these two genes is remains relatively stable and consistent under control or baseline conditions and during various experimental treatments used in our studies.

Section 4.8.: please provide the catalog numbers of used antibodies;

**Response:  In Materials/Methods-section 4.8, we have included the catalogue or product numbers for the primary antibodies and secondary antibodies used for immunoblot analysis:  

“Primary antibodies against β-actin (Cat. No. A4700, Sigma-Aldrich, St. Louis, MO), HO-1 (Cat. No. ADI-OSA-150, Enzo Life Sciences Farmingdale, NY) and Hsp70 (Cat. No. ADI-SPA-812, Enzo Life Sciences Farmingdale, NY) were used at 1:1000 (10% NFDM/TBS-T).  Primary antibodies against GCLC (Cat. No. ab41463, Abcam, Cambridge, MA) and GCLM (Cat. No. ab126704, Abcam, Cambridge, MA) were used at 1:500 (5% NFDM/TBS-T). AP-conjugated anti-mouse IgG secondary antibody (Cat. No. AP124A, Millipore-Sigma, Burlington, MA) or HRP-conjugated anti-rabbit IgG secondary antibody (Cat. No. ADI-SAB-300, Enzo Life Sciences, Farmingdale, NY)… “

Section 4.11: please provide the name and manufacturer of statistical software and the information on what kind of posthoc tests were used for each analysis;

**Response: The name and manufacturer of statistical software (JMP® statistical software by SAS) is now provided, as is information regarding posthoc tests were used for statistical analyses.

General remark: I would like to suggest changing the abbreviations of the studied genes to standardize nomenclature, I would suggest using the HUGO Gene Nomenclature Committee, available at https://www.genenames.org

**Response:  We thank the reviewer for this suggestion.  In the revised manuscript, we now use the standardized nomenclature for the rat genes analyzed in this study, which aligns with the HUGO Gene Nomenclature Committee.  The standardized names are used in the respective text of the Results, applicable figure legends, Discussion, Material/Method-section 4.7, and Table 5.  Following the HUGO nomenclature, we have used the following names in which the first letter is upper-case, and the remaining letters are lower-case:  for heat shock protein-70, Hsp4a; for hemeoxygenase, Hmox1, metallothionein, Mt1; GCL-catalytic subunit, Gclc; and GCL-modifier subunit, Gclm.  These names of rat genes are similar to those for the corresponding human gene.  However, the names of the human genes are written in all upper-case letters, e.g., heat shock protein-70 is written HSPA4.   

Round 2

Reviewer 1 Report

The authors addressed most of my concerns, the lack of the additional experiments required reduce the expected impact of the paper.